# The Canary's Echo: Auditing Privacy Risks of LLM-Generated Synthetic Text

**Matthieu Meeus** [1][2]  **Lukas Wutschitz** [2]  **Santiago Zanella-Béguelin** [2]  **Shruti Tople** [2]  **Reza Shokri** [2][3]

## Abstract

How much information about training samples can be leaked through synthetic data generated by Large Language Models (LLMs)? Overlooking the subtleties of information flow in synthetic data generation pipelines can lead to a false sense of privacy. In this paper, we assume an adversary has access to some synthetic data generated by a LLM. We design membership inference attacks (MIAs) that target the training data used to fine-tune the LLM that is then used to synthesize data. The significant performance of our MIA shows that synthetic data leak information about the training data. Further, we find that canaries crafted for model-based MIAs are sub-optimal for privacy auditing when only synthetic data is released. Such out-of-distribution canaries have limited influence on the model's output when prompted to generate useful, in-distribution synthetic data, which drastically reduces their effectiveness. To tackle this problem, we leverage the mechanics of auto-regressive models to design canaries with an in-distribution prefix and a high-perplexity suffix that leave detectable traces in synthetic data. This enhances the power of data-based MIAs and provides a better assessment of the privacy risks of releasing synthetic data generated by LLMs.

## 1. Introduction

Large Language Models (LLMs) can generate synthetic data that mimics human-written content through domain-specific prompts. Besides their impressive fluency, LLMs are known to memorize parts of their training data (Carlini et al., 2023) and can regurgitate exact phrases, sentences, or even longer passages when prompted adversarially (Zanella-Béguelin et al., 2020; Carlini et al., 2021; Nasr et al., 2023). This raises serious privacy concerns about unintended information leakage through synthetically generated text. In this paper, we address the critical question: to what extent does synthetic text generated by LLMs leak information about the real data it is derived from?

Prior methods to audit privacy risks insert highly vulnerable, out-of-distribution examples, *canaries* (Carlini et al., 2019), into the training data and test whether they can be identified using membership inference attacks (MIAs) (Shokri et al., 2017). Various MIAs have been proposed, typically assuming an attacker with access to the trained model or its output logits (Carlini et al., 2019; Shi et al., 2024). In the context of LLMs, MIAs often rely on analyzing the model's behavior when prompted with inputs related to the canaries (Carlini et al., 2021; Chang et al., 2024; Shi et al., 2024). However, similar investigations are lacking in scenarios where LLMs are used to generate synthetic data and only this synthetic data is available to an attacker.

**Contributions.** In this work, we study–for the first time–the factors that influence information leakage from a synthetic data-corpus generated using LLMs. First, we introduce data-based attacks that only have access to synthetic data, and not to the model used to generate it, and therefore cannot probe it with adversarial prompts nor compute losses or other statistics used in model-based attacks (Ye et al., 2022). We propose approximating membership likelihood using either a model trained on the synthetic data or the target example similarity to its closest synthetic data examples. We design our attacks adapting the state-of-the-art pairwise likelihood ratio tests as in RMIA (Zarifzadeh et al., 2024) and evaluate them on labeled datasets: SST-2 (Socher et al., 2013), AG News (Zhang et al., 2015) and SNLI (Bowman et al., 2015). Our results show that MIAs leveraging only synthetic data achieve AUC scores of $0.74$ for SST-2, $0.68$ for AG News and $0.77$ for SNLI, largely outperforming a random guess baseline. This suggests that synthetic text can leak significant amount of information about the real data used to generate it.

Second, we use the attacks we introduce to quantify the gap in performance between data- and model-based attacks. We do so in an auditing scenario, designing adversarial canaries and controlling leakage by varying the number of times a canary occurs in the training dataset. Experimentally, we

---

[1]Imperial College London [2]Microsoft [3]National University of Singapore. Correspondence to: Matthieu Meeus <mm422@ic.ac.uk>.

*Proceedings of the 42nd International Conference on Machine Learning*, Vancouver, Canada. PMLR 267, 2025.

find a sizable gap when comparing attacks adapted to the idiosyncrasies of each setting: a canary would need to occur $8\times$ more often to be as vulnerable against a data-based attack as it is against a model-based attack (see Figs. 1a and 1d).

Third, we discover that canaries designed for model-based attacks fall short when auditing privacy risks of synthetic text. Indeed, privacy auditing of LLMs through model-based MIAs relies on rare, out-of-distribution sequences of high perplexity (Carlini et al., 2019; Stock et al., 2022; Wei et al., 2024; Meeus et al., 2024c). We confirm that model-based MIAs improve as canary perplexity increases. In sharp contrast, we find that high perplexity sequences, although distinctly memorized by the target model, are less likely to be *echoed* through synthetic data generated by the target model. Therefore, as a canary perplexity increases, the canary influence on synthetic data decreases, making its membership less detectable from synthetic data (see Figure 2). We show that low-perplexity, and even in-distribution canaries, while suboptimal for model-based attacks, are more adequate canaries in data-based attacks.

Next, we propose an alternative canary design tailored for data-based attacks based on the following intuition: (i) in-distribution canaries aligned with the domain-specific prompt can influence the generated output; and (ii) memorization is more likely when canaries contain sub-sequences with high perplexity. We construct canaries starting with an in-distribution prefix of length $F$, transitioning into an out-of-distribution suffix, increasing the likelihood that the model memorizes them and that they influence synthetic data. We show that, for fixed overall canary perplexity, the performance of attacks for canaries with in-distribution prefix and out-of-distribution suffix ($0 < F < \max$) improves upon both entirely in-distribution canaries ($F = \max$) and out-of-distribution canaries ($F = 0$), across datasets (see Fig. 1 and Table 2).

Lastly, we evaluate our attacks on synthetic data generated with formal privacy guarantees. We adopt the training-time method proposed by work (Yue et al., 2023; Mattern et al., 2022; Kurakin et al., 2023) and finetune the target model on the private dataset using DP-SGD (Abadi et al., 2016) with $\epsilon = 8$. We find the performance of the strongest data-based MIA to drop to random guess performance (AUC of 0.5), confirming that differential privacy constitutes a strong defense.

Taken together, the proposed MIAs and canary design can be used to audit privacy risks of synthetic text. Auditing establishes a lower bound on the risk, useful to take informed decisions about releasing synthetic data in sensitive applications and also complements upper bounds on privacy risks from methods that synthesize text with provable guarantees.

## 2. Background and problem statement

**Synthetic text generation.** We consider a private dataset $D = \{x_i = (s_i, \ell_i)\}_{i=1}^N$ of labelled text records where $s_i$ represents a sequence of tokens (e.g. a product review) and $\ell_i$ is a class label (e.g. the review sentiment). A synthetic data generation mechanism is a probabilistic procedure mapping $D$ to a synthetic dataset $\widetilde{D} = \{\widetilde{x}_i = (\widetilde{s}_i, \widetilde{\ell}_i)\}_{i=1}^{\widetilde{N}}$ with a desired label set $\{\ell_i\}_{i=1}^{\widetilde{N}}$. Unless stated otherwise, we consider $N = \widetilde{N}$. The synthetic dataset $\widetilde{D}$ should preserve the *utility* of the private dataset $D$, i.e., it should preserve as many statistics of $D$ that are useful for downstream analyses as possible. In addition, a synthetic data generation mechanism should preserve the *privacy* of records in $D$, i.e. it should not leak sensitive information from the private records into the synthetic records. The utility of a synthetic dataset can be measured by the gap between the utility achieved by $\widetilde{D}$ and $D$ in downstream applications. The fact that synthetic data is not *directly* traceable to original data records does not mean that it is free from privacy risks. On the contrary, the design of a synthetic data generation mechanism determines how much information from $D$ leaks into $\widetilde{D}$ and should be carefully considered. Indeed, several approaches have been proposed to generate synthetic data with formal privacy guarantees (Kim et al., 2021; Tang et al., 2024; Wu et al., 2024; Xie et al., 2024). We focus on privacy risks of text generated by a pre-trained LLM fine-tuned on a private dataset $D$ (Yue et al., 2023; Mattern et al., 2022; Kurakin et al., 2023). Specifically, we fine-tune an LLM $\theta_0$ on records $(s_i, \ell_i) \in D$ to minimize the loss in completing $s_i$ conditioned on a prompt template $\mathsf{p}(\ell_i)$, obtaining $\theta$. We then query $\theta$ using the same prompt template to build a synthetic dataset $\widetilde{D}$ matching a given label distribution.

**Membership inference attacks.** MIAs (Shokri et al., 2017) provide a meaningful measure to quantify privacy risks of machine learning models, due to its simplicity but also due to the fact that protection against MIAs implies protection against more devastating attacks such as attribute inference and data reconstruction (Salem et al., 2023). In a MIA on a target model $\theta$, an adversary aims to infer whether a target record is present in the training dataset of $\theta$. Different variants constrain the adversary's access to the model. In our setting, we consider model-based adversaries that observe the output logits on inputs of their choosing of a model $\theta$ fine-tuned on a private dataset $D$. We naturally extend the concept of MIAs to synthetic data generation mechanisms by considering data-based adversaries that only observe a synthetic dataset $\widetilde{D}$ generated from $D$.

**Privacy auditing using canaries.** A common method used to audit the privacy risks of ML models is to evaluate the MIA vulnerability of canaries, i.e., artificial worst-case records inserted in otherwise natural datasets (Carlini et al., 2019). This method can also be employed to de-

rive statistical lower bounds on the differential privacy (DP) guarantees of the training pipeline (Jagielski et al., 2020; Zanella-Béguelin et al., 2023). Records crafted to be out-of-distribution w.r.t. the underlying data distribution of $D$ give a good approximation to the worst-case (Carlini et al., 2019; Meeus et al., 2024c). Canaries can take a range of forms, such as text containing sensitive information (Carlini et al., 2019) and random (Wei et al., 2024) or synthetically generated sequences (Meeus et al., 2024c). Prior work identified that longer sequences, repeated more often (Carlini et al., 2023), and with higher perplexity (Meeus et al., 2024c) are better memorized during training and hence are more vulnerable to model-based MIAs. We study multiple types of canaries and compare their vulnerability against model- and synthetic data-based MIAs. We consider a set of canaries $\{\hat{x}_i = (\hat{s}_i, \hat{\ell}_i)\}_{i=1}^{\hat{N}}$, each crafted adversarially and inserted with probability ½ into the private dataset $D$. The resulting dataset is then fed to a synthetic data generation mechanism. We finally consider each canary $\hat{x}_i$ as the target record of a MIA to estimate the privacy risk of the generation mechanism (or the underlying fine-tuned model).

**Threat model.** We consider an adversary $\mathcal{A}$ who aims to infer whether a canary $\hat{x}$ was included in the private dataset $D$ used to synthesize a dataset $\widetilde{D}$. We distinguish between two threat models: (i) an adversary $\mathcal{A}^\theta$ with query-access to output logits of a target model $\theta$ fine-tuned on $D$; and (ii) an adversary $\mathcal{A}^{\widetilde{D}}$ with only access to the synthetic dataset $\widetilde{D}$. To the best of our knowledge, for text data this latter threat model has not been studied extensively in the literature. In contrast, the privacy risks of releasing synthetic tabular data are much better understood (Stadler et al., 2022; Yale et al., 2019; Hyeong et al., 2022; Zhang et al., 2022). Algorithm 1 shows the generic membership inference experiment encompassing both model- and data-based attacks, selected by the synthetic flag. The adversary is represented by a stateful procedure $\mathcal{A}$, used to craft a canary and compute its membership score. Compared to a standard membership experiment, we consider a fixed private dataset $D$ rather than sampling it, and let the adversary choose the target $\hat{x}$. This is close to the threat model of *unbounded* DP, where the implicit adversary selects two datasets, one obtained from the other by adding one more record, except that in our case the adversary observes but cannot choose the records in $D$. The membership score $\beta$ returned by the adversary can be turned into a binary membership label by choosing an appropriate threshold. We further clarify assumptions made for the adversary in both threat models in Appendix E.

**Problem statement.** We study methods to audit privacy risks associated with releasing synthetic text. Our main goal is to develop an effective data-based adversary $\mathcal{A}^{\widetilde{D}}$ in the threat model of Algorithm 1. For this, we explore the design space of canaries to approximate the worst-case, and

adapt state-of-the-art methods used to compute membership scores in model-based attacks to the data-based scenario.

## 3. Methodology

### 3.1. Computing the membership score

In Algorithm 1, the adversary computes a membership score $\beta$ indicating their confidence that $\theta$ was trained on $\hat{x}$ (i.e. that $b = 1$). We specify first how to compute a membership signal $\alpha$ for model- and data-based adversaries, and then how we compute $\beta$ from $\alpha$ adapting the RMIA methodology of Zarifzadeh et al. (2024).

#### 3.1.1. MODEL-BASED ATTACKS

The larger the target model $\theta$'s probability for canary $\hat{x} = (\hat{s}, \hat{\ell})$, $P_\theta(\hat{s} \mid \mathsf{p}(\hat{\ell}))$, as compared to its probability on reference models, the more likely that the model has seen this record during training. We compute the probability for canary $\hat{x}$ as the product of token-level probabilities for $\hat{s}$ conditioned on the prompt $\mathsf{p}(\hat{\ell})$. Given a target canary text $\hat{s} = t_1, \ldots, t_n$, we compute $P_\theta(\hat{s} \mid \mathsf{p}(\hat{\ell}))$ as $P_\theta(\hat{x}) = \prod_{j=1}^n P_\theta(t_j \mid \mathsf{p}(\hat{\ell}), t_1, \ldots, t_{j-1})$. We consider this probability as the membership inference signal against a model, i.e. $\alpha = P_\theta(\hat{s} \mid \mathsf{p}(\hat{\ell}))$.

#### 3.1.2. DATA-BASED ATTACKS

When the attacker only has access to the synthetic data $\widetilde{D}$, we need to extract a signal purely from $\widetilde{D}$ that correlates with membership. We next describe two methods to compute a membership signal $\alpha$ based on $\widetilde{D}$. For more details, refer to their pseudo-code in Appendix A.

$n$-**gram model.** The attacker first fits an $n$-gram model using $\widetilde{D}$ as training corpus. An $n$-gram model computes the probability of the next token $w_j$ in a sequence based solely on the previous $n - 1$ tokens (Jurafsky & Martin, 2024). The conditional probability of a token $w_j$ given the previous $n - 1$ tokens is estimated from the counts of $n$-grams in the training corpus. Formally, $P_{n\text{-gram}}(w_j \mid w_{j-(n-1)}, \ldots, w_{j-1}) = \frac{C(w_{j-(n-1)}, \ldots, w_j)+1}{C(w_{j-(n-1)}, \ldots, w_{j-1})+V}$, where $C(s)$ is the number of times the sequence $s$ appears in the training corpus and $V$ is the vocabulary size. We use Laplace smoothing to deal with $n$-grams that do not appear in the training corpus, incrementing the count of every $n$-gram by 1. The probability that the model assigns to a sequence of tokens $s = (w_1, \ldots, w_k)$ can be computed as $P_{n\text{-gram}}(s) = \prod_{j=2}^k P_{n\text{-gram}}(w_j \mid w_{j-(n-1)}, \ldots, w_{j-1})$. With the $n$-gram model fitted on the synthetic dataset, the attacker computes the $n$-gram model probability of the target canary $\hat{x} = (\hat{s}, \hat{\ell})$ as its membership signal, i.e. $\alpha = P_{n\text{-gram}}(\hat{s})$. Intuitively, if the canary $\hat{x}$ was present in the training data, the generated synthetic data $\widetilde{D}$ will bet-

---

**Algorithm 1** Membership inference against an LLM-based synthetic text generator

---

1: **Input**: Fine-tuning algorithm $\mathcal{T}$, pre-trained model $\theta_0$, private dataset $D = \{x_i = (s_i, \ell_i)\}_{i=1}^{N}$, labels $\{\widetilde{\ell}_i\}_{i=1}^{\widetilde{N}}$, prompt template $\mathsf{p}(\cdot)$, canary repetitions $n_{\text{rep}}$, sampling method $\mathsf{sample}$, adversary $\mathcal{A}$
2: **Output**: Membership score $\beta$
3: $\hat{x} \leftarrow \mathcal{A}(\mathcal{T}, \theta_0, D, \{\widetilde{\ell}_i\}_{i=1}^{\widetilde{N}}, \mathsf{p}(\cdot))$          {Adversarially craft a canary (see Sec. 3.2)}
4: $b \sim \{0, 1\}$          {Flip a fair coin}
5: **if** b = 1 **then**
6:     $\theta \leftarrow \mathcal{T}(\theta_0, D \cup \{\hat{x}\}^{n_{\text{rep}}})$          {Fine-tune $\theta_0$ with canary repeated $n_{\text{rep}}$ times}
7: **else**
8:     $\theta \leftarrow \mathcal{T}(\theta_0, D)$          {Fine-tune $\theta_0$ without canary}
9: **end if**
10: **for** $i = 1 \ldots \widetilde{N}$ **do**
11:     $\widetilde{s}_i \sim \mathsf{sample}(\theta(\mathsf{p}(\widetilde{\ell}_i)))$          {Sample synthetic records using prompt template}
12: **end for**
13: $\widetilde{D} \leftarrow \left\{ (\widetilde{s}_i, \widetilde{\ell}_i) \right\}_{i=1}^{\widetilde{N}}$
14: **if** synthetic **then**
15:     $\beta \leftarrow \mathcal{A}(\widetilde{D}, \hat{x})$          {Compute membership score $\beta$ of $\hat{x}$, see Sec. 3.1.2 and algorithms in Appendix A}
16: **else**
17:     $\beta \leftarrow \mathcal{A}(\theta, \hat{x})$          {Compute membership score $\beta$ of $\hat{x}$, see Sec. 3.1.1}
18: **end if**
19: **return** $\beta$

---

ter reflect the patterns of $\hat{s}$, resulting in the $n$-gram model assigning a higher probability to $\hat{s}$ than if it was not present.

**Similarity metric.** The attacker computes the similarity between the target canary text $\hat{s}$ and all synthetic sequences $\widetilde{s}_i$ in $\widetilde{D}$ using similarity metric SIM, i.e. $\sigma_i = \text{SIM}(\hat{s}, \widetilde{s}_i)$ for $i = 1, \ldots, \widetilde{N}$. Next, the attacker identifies the $k$ synthetic sequences with the largest similarity to $\hat{s}$. With $\sigma_{i(j)}$ the $j$-th largest similarity, the membership inference signal is computed as the mean of the $k$ most similar examples, i.e. $\alpha = \frac{1}{k} \sum_{j=1}^{k} \sigma_{i(j)}$. Intuitively, if $\hat{s}$ was part of the training data, the synthetic data $\widetilde{D}$ will likely contain sequences $\widetilde{s}_i$ more similar to $\hat{s}$ than if $\hat{s}$ was not part of the training data, resulting in a larger mean similarity. Various similarity metrics can be used. We consider Jaccard similarity ($\text{SIM}_{\text{Jac}}$), often used to measure string similarity, and cosine similarity between the embeddings of the two sequences, computed using a pre-trained embedding model ($\text{SIM}_{\text{emb}}$).

### 3.1.3. COMPUTING RMIA SCORES

Reference models, also called *shadow* models, are surrogate models designed to approximate the behavior of a target model. MIAs based on reference models perform better but are more costly to run than MIAs that do not use them, with the additional practical challenge that they require access to data distributed similarly to the training data of the target model (Shokri et al., 2017; Ye et al., 2022). Obtaining multiple reference models in our scenario requires fine-tuning a large number of parameters in an LLM and

quickly becomes computationally prohibitive. We use the state-of-the-art RMIA method (Zarifzadeh et al., 2024) to maximize attack performance with a limited number of reference models $M$. Specifically, for the target model $\theta$, we calculate the membership score of a canary $\hat{x}$ using reference models $\{\theta_i'\}_{i=1}^{M}$ as follows (details on applying RMIA to our setup are in Appendix B): $\beta_\theta(\hat{x}) = \frac{\alpha_\theta(\hat{x})}{\frac{1}{M} \sum_{i=1}^{M} \alpha_{\theta_i'}(\hat{x})}$.

### 3.2. Canary generation

Prior work has shown that canaries with high perplexity are more likely to be memorized by language models (Meeus et al., 2024c). High perplexity sequences are less predictable and require the model to encode more specific, non-generalizable details about them. However, high perplexity canaries are not necessarily more susceptible to leakage via synthetic data generation, as they are outliers in the text distribution when conditioned on a given in-distribution prompt. This misalignment with the model's natural generative behavior means that even when memorized, these canaries are unlikely to be reproduced during regular model inference, making them ineffective for detecting memorization of training examples in generated synthetic data.

To address this issue, we take advantage of the greedy nature of popular autoregressive decoding strategies (e.g. beam search, top-$k$ and top-$p$ sampling). We can encourage such decoding strategies to generate text closer to canaries by crafting canaries with a low perplexity prefix. To ensure memorization, we follow established practices and choose

a high perplexity suffix. Specifically, we design canaries $\hat{x} = (\hat{s}, \hat{\ell})$, where $\hat{s}$ has an **in-distribution prefix** and an **out-of-distribution suffix**. In practice, we split the original dataset $D$ into a training dataset and a canary source dataset. For each record $x = (s, \ell)$ in the canary source dataset, we design a new canary $\hat{x} = (\hat{s}, \hat{\ell})$. We truncate $s$ to get an in-distribution prefix of length $F$ and generate a suffix using the pre-trained language model $\theta_0$, adjusting the sampling temperature to achieve a desired target perplexity $\mathcal{P}_{\text{target}}$. We use rejection sampling to ensure that the perplexity of the generated canaries falls within the range $[0.9\,\mathcal{P}_{\text{target}}, 1.1\,\mathcal{P}_{\text{target}}]$. We ensure the length is consistent across canaries, as this impacts memorization (Carlini et al., 2023; Kandpal et al., 2022). By adjusting the length of the in-distribution prefix, we can guide the generation of either entirely in-distribution or out-of-distribution canaries.

We insert each canary $n_{\text{rep}}$ times in the training dataset of target and reference models. When a canary is selected as a *member*, the canary is repeated $n_{\text{rep}}$ times in the training dataset, while canaries selected as *non-members* are excluded from the training dataset. As in prior work (Carlini et al., 2023; Kandpal et al., 2022; Meeus et al., 2024c), we opt for $n_{\text{rep}} > 1$ to increase memorization, thus facilitating privacy auditing and the observation of the effect of different factors on the performance of MIAs during ablation studies.

## 4. Experimental setup

**Datasets.** We consider three datasets that have been widely used to study text classification: (i) the Stanford Sentiment Treebank (**SST-2**) (Socher et al., 2013), which consists of excerpts from written movie reviews with a binary sentiment label; (ii) the **AG News** dataset (Zhang et al., 2015), which consists of news articles labelled by category (World, Sport, Business, Sci/Tech).; and (iii) the **SNLI** dataset (Bowman et al., 2015), which consists of premises and hypotheses labeled as entailment, contradiction or neutral. In all experiments, we remove examples with less than 5 words, bringing the total number of examples to 43 296 for SST-2 and 120 000 for AG News. For SNLI, we selected the first 100 000 records.

**Synthetic data generation.** We fine-tune the pre-trained Mistral-7B model (Jiang et al., 2023) using low-rank adaptation (LoRA) (Hu et al., 2022). We use a custom prompt template $\mathsf{p}(\cdot)$ for each dataset (see Appendix C). More details on the implementation and parameters are provided in Appendix D. We sample synthetic data from the fine-tuned model $\theta$ conditioned on prompts $\mathsf{p}(\widetilde{\ell}_i)$, following the same distribution of labels in the synthetic dataset $\widetilde{D}$ as in the original dataset $D$, i.e. $\ell_i = \widetilde{\ell}_i$ for $i = 1, ..., \widetilde{N}$. To generate synthetic sequences, we sequentially sample completions using a softmax temperature of 1.0 and top-$p$ (aka nucleus)

sampling with $p = 0.95$, i.e. we sample from a vocabulary restricted to the smallest possible set of tokens whose total probability exceeds 0.95. We further ensure that the synthetic data bears high utility, and is thus realistic. For this, we consider the downstream classification tasks for which the original datasets have been designed. We fine-tune RoBERTa-base (Liu et al., 2019) on $D$ and $\widetilde{D}$ and compare the performance of the resulting classifiers on held-out evaluation datasets. Details are provided in Appendix F, for synthetic data generated with and without canaries.

**Canary injection.** We generate canaries $\hat{x} = (\hat{s}, \hat{\ell})$ as described in Sec. 3.2. Unless stated otherwise, we consider 50-word canaries. Synthetic canaries are generated using Mistral-7B (Jiang et al., 2023) as $\theta_0$. We consider two ways of constructing a canary label: (i) randomly sampling a label $\hat{\ell}$ from the distribution of labels in $D$, ensuring that the class distribution among canaries matches that of $D$ (*Natural*); and (ii) extending the set of labels with a new artificial label ($\hat{\ell} =$ "canary") only used for canaries (*Artificial*).

**Membership inference.** We compute the membership scores $\beta_\theta(\hat{x})$ as described in Sec. 3.1. For one target model $\theta$, we consider 1000 canaries $\hat{x}$, of which on average half are included in the training dataset $n_{\text{rep}}$ times (members), while the remaining half are excluded (non-members). We then use the computed RMIA scores and the ground truth for membership to construct ROC curves, from which we compute AUC and true positive rate (TPR) at low false positive rate (FPR) as measures of MIA performance. Across experiments, we use $M = 4$ reference models $\theta'$, each trained on a dataset $D_{\theta'}$ consisting of the dataset $D$ used to train the target model $\theta$ with canaries inserted. Note that although practical attacks rarely have this amount of information, this is allowed by the threat model of Algorithm 1 and valid as a worst-case auditing methodology. We ensure that each canary is a member in half (i.e. 2) of the reference models and a non-member in the other half. For the attacks based on synthetic data, we use $n = 2$ for computing scores using an $n$-gram model and $k = 25$ for computing scores based on similarity. We use Sentence-BERT (Reimers & Gurevych, 2019) (`paraphrase-MiniLM-L6-v2` from `sentence-transformers`) as the embedding model.

## 5. Results

### 5.1. Baseline evaluation with standard canaries

We begin by assessing the vulnerability of synthetic text using standard canaries. Specifically, we utilize both in-distribution canaries and synthetically generated canaries with a target perplexity $\mathcal{P}_{\text{target}} = 250$, no in-distribution prefix ($F = 0$), $n_{\text{rep}} = 12$ and *natural* or *artificial* labels, as described in Section 4. Table 1 summarizes the ROC AUC for model- and data-based attacks.

| Dataset | Canary injection | | Model $\mathcal{A}^\theta$ | Synthetic $\mathcal{A}^{\widetilde{D}}$ (2-gram) | Synthetic $\mathcal{A}^{\widetilde{D}}$ (SIM$_{\text{Jac}}$) | Synthetic $\mathcal{A}^{\widetilde{D}}$ (SIM$_{\text{emb}}$) |
| | Source | Label | | | | |
|---|---|---|---|---|---|---|
| SST-2 | In-distribution[1] | | 0.911 | 0.741 | 0.602 | 0.586 |
| | Synthetic | Natural | 0.999 | 0.620 | 0.547 | 0.530 |
| | | Artificial | 0.999 | 0.682 | 0.552 | 0.539 |
| AG News | In-distribution | | 0.993 | 0.676 | 0.590 | 0.565 |
| | Synthetic | Natural | 0.996 | 0.654 | 0.552 | 0.506 |
| | | Artificial | 0.999 | 0.672 | 0.560 | 0.525 |
| SNLI | In-distribution[1] | | 0.892 | 0.718 | 0.644 | 0.630 |
| | Synthetic | Natural | 0.998 | 0.534 | 0.486 | 0.488 |
| | | Artificial | 0.997 | 0.770 | 0.602 | 0.571 |

[1] Constrained by in-distribution data, canaries consist of exactly 30 words (50 elsewhere).

*Table 1.* ROC AUC across datasets, threat models (model-based $\mathcal{A}^\theta$ and data-based $\mathcal{A}^{\widetilde{D}}$) and MIA methodologies for standard, high perplexity canaries (target perplexity $\mathcal{P}_{\text{target}} = 250$, no in-distribution prefix ($F = 0$) and $n_{\text{rep}} = 12$). We give the ROC curves and TPR at low FPR scores in Appendix G, further ablations in Appendix H, and elaborate on the disparate vulnerability of high perplexity canaries in model- and data-based attacks in Appendix I.

First, we find that MIAs relying solely on the generated synthetic data achieve a AUC score significantly higher than a random guess (i.e. AUC = 0.5), reaching up to 0.74 for SST-2, 0.68 for AG News and 0.77 for SNLI. This shows that synthetic text can leak information about the real data used to generate it.

Next, we observe that the data-based attack using an $n$-gram model trained on synthetic data to compute membership scores outperforms the two attacks leveraging similarity metrics: Jaccard distance between a canary and synthetic strings (SIM$_{\text{Jac}}$) or cosine distance between their embeddings (SIM$_{\text{emb}}$). This suggests that information critical to infer membership lies in subtle changes in the co-occurrence of $n$-grams in synthetic data rather than in the generation of many sequences with lexical or semantic similarity.

We also compare MIA performance across different canary types under data-based attacks. The AUC remains consistently higher than a random guess across all canaries. For SST-2 and AG News, the highest AUC score of 0.74 and 0.68 is achieved when using in-distribution canaries, while for SNLI the AUC of 0.77 is reached for synthetic canaries.

As another baseline, we test RMIA on the target model trained on $D$, assuming the attacker has access to the model logits ($\mathcal{A}^\theta$). This attack achieves near-perfect performance across all setups, highlighting an inherent gap between the performance of model- and data-based MIAs. This suggests that, while a fine-tuned model memorizes standard canaries well, the information necessary to infer their membership is only partially transmitted to the synthetic text.

To investigate the gap between the two attacks in more detail, we vary the number of canary repetitions $n_{\text{rep}}$ to amplify the power of the data-based attack until its performance

matches that of a model-based attack. Fig. 1a illustrates these results as a set of ROC curves. We quantify this discrepancy by noting that the MIA performance for $\mathcal{A}^{\widetilde{D}}$ at $n_{\text{rep}} = 16$ is comparable to $\mathcal{A}^\theta$ at $n_{\text{rep}} = 2$ and for low FPR at $n_{\text{rep}} = 1$. We find similar results in Fig. 1d for AG News. The MIA performance for $\mathcal{A}^{\widetilde{D}}$ at $n_{\text{rep}} = 16$ falls between the performance of $\mathcal{A}^\theta$ at $n_{\text{rep}} = 1$ and $n_{\text{rep}} = 2$. Under these experimental conditions, canaries would need to be repeated 8 to 16× to reach the same vulnerability in data-based attacks compared to model-based attacks.

We provide additional results for the standard canaries as appendices: TPR at low FPR scores in Appendix G, ablations for data-based MIA hyperparameters in Appendix H, and a discussion on the disparate vulnerability of high perplexity canaries in model- and data-based attacks in Appendix I.

### 5.2. Specialized canaries for enhanced privacy auditing

To effectively audit privacy risks in a worst-case scenario, we explore designing specialized canaries that are both memorized by the model and influential in the synthetic data.

First, we generate specialized canaries by controlling their target perplexity $\mathcal{P}_{\text{target}}$. We evaluate MIAs for both threat models across a range of perplexities for canaries with natural labels, using $n_{\text{rep}} = 4$ for the model-based MIA $\mathcal{A}^\theta$ and $n_{\text{rep}} = 16$ for the data-based MIA $\mathcal{A}^{\widetilde{D}}$. We explore a wide range of perplexities, finding $1 \times 10^5$ to align with random token sequences. Figure 2 shows the ROC AUC score versus canary perplexity. For the model-based attack $\mathcal{A}^\theta$, the AUC monotonically increases with canary perplexity, reaffirming that outlier records with higher perplexity are more vulnerable to MIAs (Feldman & Zhang, 2020; Carlini et al., 2022a;

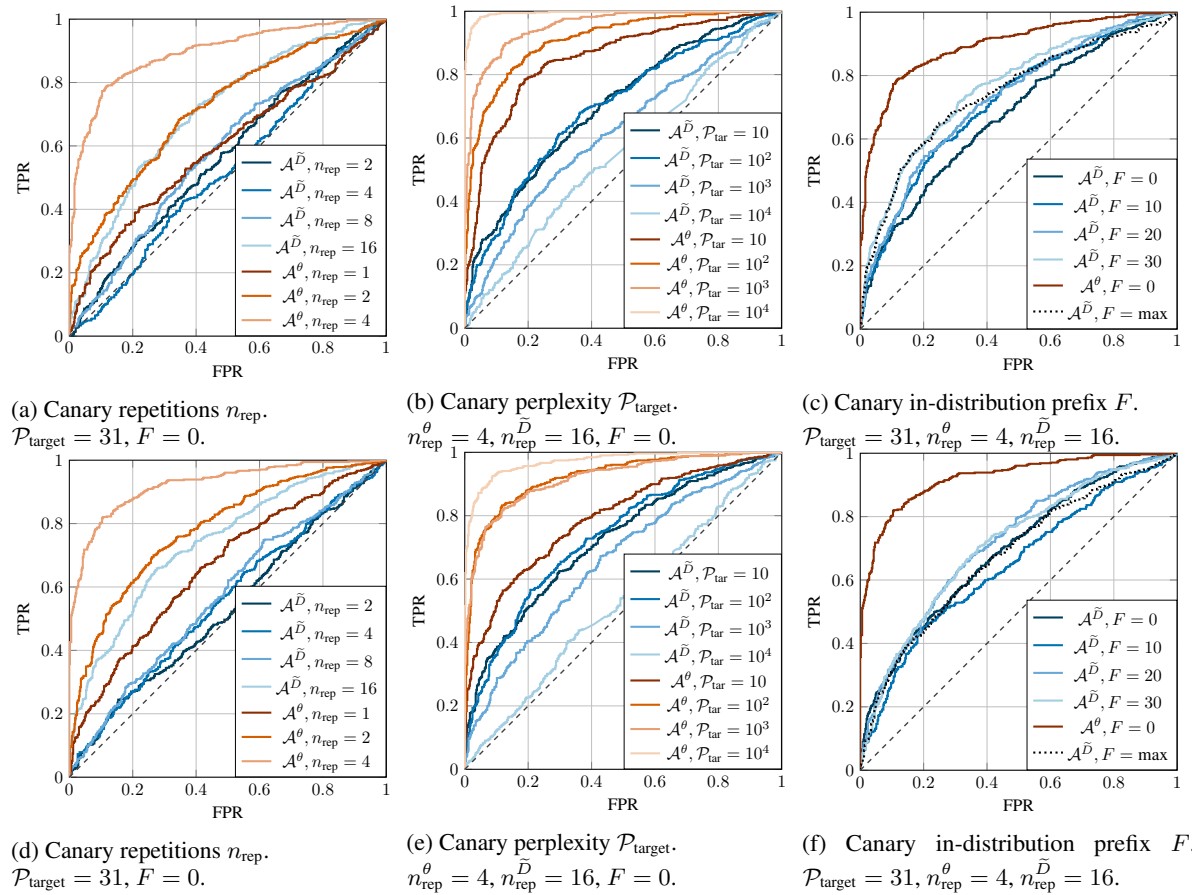

**(a)** Canary repetitions $n_{\text{rep}}$. $\mathcal{P}_{\text{target}} = 31$, $F = 0$.

**(b)** Canary perplexity $\mathcal{P}_{\text{target}}$. $n_{\text{rep}}^{\theta} = 4$, $n_{\text{rep}}^{\widetilde{D}} = 16$, $F = 0$.

**(c)** Canary in-distribution prefix $F$. $\mathcal{P}_{\text{target}} = 31$, $n_{\text{rep}}^{\theta} = 4$, $n_{\text{rep}}^{\widetilde{D}} = 16$.

**(d)** Canary repetitions $n_{\text{rep}}$. $\mathcal{P}_{\text{target}} = 31$, $F = 0$.

**(e)** Canary perplexity $\mathcal{P}_{\text{target}}$. $n_{\text{rep}}^{\theta} = 4$, $n_{\text{rep}}^{\widetilde{D}} = 16$, $F = 0$.

**(f)** Canary in-distribution prefix $F$. $\mathcal{P}_{\text{target}} = 31$, $n_{\text{rep}}^{\theta} = 4$, $n_{\text{rep}}^{\widetilde{D}} = 16$.

*Figure 1.* ROC curves of MIAs on synthetic data $\mathcal{A}^{\widetilde{D}}$ compared to model-based MIAs $\mathcal{A}^{\theta}$ on SST-2 (1a–1c) and AG News (1d–1f). We ablate over the number of canary insertions $n_{\text{rep}}$ in 1a, 1d, the target perplexity $\mathcal{P}_{\text{target}}$ of the inserted canaries in 1b, 1e and the length $F$ of the in-distribution prefix in the canary in 1c, 1f. Log-log plots in Appendix J.

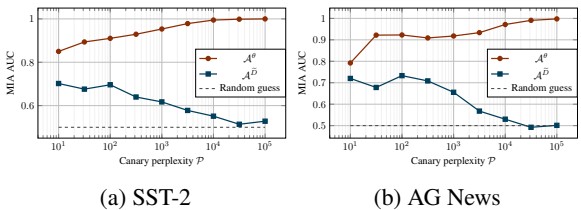

**(a)** SST-2     **(b)** AG News

*Figure 2.* ROC AUC for synthetic canaries with varying perplexity (natural label). The model-based MIA $\mathcal{A}^{\theta}$ improves as canary perplexity increases, while the data-based MIA performance $\mathcal{A}^{\widetilde{D}}$ (2-gram) decreases. $n_{\text{rep}}^{\theta} = 4$, $n_{\text{rep}}^{\widetilde{D}} = 16$.

Meeus et al., 2024c). Conversely, for the data-based attack $\mathcal{A}^{\widetilde{D}}$, the AUC initially increases with perplexity but starts to decline beyond a certain threshold, eventually approaching a random guess (AUC of 0.5). To further illustrate this, we present the complete ROC curve in Figures 1b and 1e for SST-2 and AG News, respectively. We vary the canary perplexity $\mathcal{P}_{\text{target}}$ while keeping other parameters constant. As $\mathcal{P}_{\text{target}}$ increases, the model-based attack improves across

the entire FPR range, while the data-based attack weakens, approaching AUC of 0.5 at high perplexities. This suggests that identifying susceptible canaries is straightforward for model-based privacy audits, but assessing the privacy risk of synthetic data requires a careful balance between canary memorization and its influence on synthetic data.

We now examine whether canaries can be crafted to enhance both memorization and influence on the synthetic data, making them suitable to audit the privacy risks of releasing synthetic data. In Sec. 3.2, we introduced a method that exploits the greedy nature of LLM decoding to design more vulnerable canaries. We craft a canary with a low-perplexity, in-distribution prefix to optimize its impact on the synthetic dataset, followed by a high-perplexity suffix to enhance memorization. We generate this suffix sampling from the pre-trained LLM $\theta_0$ with high temperature. Figures 1c and 1f illustrate the ROC curves for SST-2 and AG News, respectively, and Table 2 summarizes the corresponding ROC AUC and TPR at low FPR. We set the overall canary perplexity $\mathcal{P}_{\text{target}} = 31$ and vary the prefix length $F$ from $F = 0$ (fully

| Dataset | F | ROC AUC | TPR@ | |
| | | | FPR=0.01 | FPR=0.1 |
|---|---|---|---|---|
| SST-2 | 0 | 0.673 | 0.081 | 0.304 |
| | 10 | 0.715 | 0.057 | 0.312 |
| | 20 | 0.725 | 0.069 | 0.318 |
| | 30 | **0.760** | 0.069 | **0.410** |
| | max | 0.741 | **0.101** | 0.408 |
| AG News | 0 | 0.692 | **0.089** | 0.309 |
| | 10 | 0.646 | 0.053 | 0.276 |
| | 20 | **0.716** | 0.069 | 0.321 |
| | 30 | 0.710 | 0.055 | **0.333** |
| | max | 0.676 | 0.039 | 0.314 |

*Table 2.* MIA performance (ROC AUC and TPR at low FPR) for data-based MIA $\mathcal{A}^{\bar{D}}$ (2-gram) for canaries with varying length of in-distribution prefix $F$ (results from Figs. 1c,1f).

synthetic canaries) to $F = $ max (in-distribution canaries). We observe that combining an in-distribution prefix ($F > 0$) with a high-perplexity suffix ($F < $ max) enhances attack effectiveness. For both datasets, the optimal AUC, and often also the optimal TPR at low FPR, for the MIA is reached for a prefix length $0 < F < $ max (see Table 2). This suggests that although the model's memorization of the canary stays consistent (as the overall perplexity remains unchanged), the canary's impact on the synthetic data becomes more prominent with longer in-distribution prefixes. We hypothesize that familiar low-perplexity prefixes serve as starting points for text generation, enhancing the likelihood that traces of the canary appear in the synthetic data.

### 5.3. Identifying the memorized sub-sequences

We analyze what information from a canary leaks into the synthetic data that enables a data-based attack to infer its membership. For each canary $\hat{x} = (\hat{s}, \hat{\ell})$, we examine the synthetic data generated by a model trained on a dataset including (member) and excluding $\hat{x}$ (non-member). We leverage the $M = 4$ reference models $\theta'$ used to develop the attack for 1000 specialized canaries from Fig. 1c. For each model $\theta'$, we count the number of $n$-grams in $\widetilde{s}$ that occur at least once in $\widetilde{D}'$ ($C_{\text{unique}}$). We also compute the median $C_{\text{med}}$ and average $C_{\text{avg}}$ counts of $n$-grams from $\hat{s}$ in $\widetilde{D}'$. Table 3 summarizes how these measures vary with $n$. As $n$ increases, the number of $n$-grams from the canary appearing in the synthetic data drops sharply, reaching $C_{\text{med}} = 0$ for $n = 4$ for models including and excluding a canary. This suggests that any verbatim reproduction of canary text in the generated synthetic data is of limited length. Further, we observe only slight differences in counts between members and non-members, indicating that the signal for inferring membership is likely in subtle shifts in the probability distribution of token co-occurrences within the synthetic data, as captured by the 2-gram model. We further analyze canaries with the highest and lowest RMIA scores in Appendix K.

### 5.4. Synthetic data with formal privacy guarantees

To mitigate any privacy leakage associated with the release of synthetic data, prior work has proposed to generate synthetic data with formal privacy guarantees, in particular differential privacy (DP). Methods used to generate synthetic text with DP guarantees mitigate MIAs by ensuring that any single training record exerts limited influence on synthesized data. These methods are broadly split into training-time (Yue et al., 2023; Mattern et al., 2022; Kurakin et al., 2023) and inference-time (Xie et al., 2024; Wu et al., 2024; Tang et al., 2024; Amin et al., 2024). Training-time methods fine-tune a pre-trained LLM with DP-SGD and then prompt this model to generate synthetic data. These methods leverage the post-processing property of DP to transfer the guarantees from the fine-tuned model to synthetic data. Because generating synthetic data from a DP model does not consume additional privacy budget, they can generate an unlimited amount of data with a fixed privacy budget. In contrast, inference-time methods use unmodified pre-trained models prompted on private data and inject calibrated noise during decoding (Xie et al., 2024; Wu et al., 2024; Tang et al., 2024) or employ DP evolutionary algorithms to steer generation towards a distribution similar to the private data (Amin et al., 2024).

We instantiate the training-time method, i.e. finetuning the target model with DP-SGD (Abadi et al., 2016) using the Opacus library (Yousefpour et al., 2021) and $\epsilon = 8$. We follow the same setup from Section 5.1 and report the performance of the data-based MIA in Table 4. As expected, we find the AUC for the strongest data-based MIA (2-gram) to approach random guess performance (AUC of 0.5) when DP guarantees are incorporated. This confirms that DP constitutes a strong defense. We further find that the corresponding generated synthetic data maintains a high utility in downstream tasks. For instance, for synthetic data generated with $\epsilon = 8$, accuracy on SST-2 reaches 91.6%, compared to 91.5% for non-DP synthetic data and 92.3% for real data (see Appendix F).

Our results suggest that DP-generated synthetic data can achieve high utility, while strongly mitigating the success of data-based MIAs. Yet, achieving the right balance between privacy and utility in DP synthetic text generation is likely context-dependent. We hope that the privacy auditing framework adapted to actual threat models we here propose enables future work to rigorously explore this trade-off.

## 6. Related work

**MIAs against ML models.** Since the seminal work of Shokri et al. (2017), MIAs have been used to study memorization and privacy risks. Model-based MIAs have been studied under varying threat models, including adversaries

| | $C_{unique}$ | | $C_{med}$ | | $C_{avg}$ | |
|---|---|---|---|---|---|---|
| $n$ | Member | Non-member | Member | Non-member | Member | Non-member |
| 1 | $45.97 \pm 2.8$ | $45.1 \pm 3.0$ | $819.6 \pm 702.6$ | $821.5 \pm 727.9$ | $7389.2 \pm 1648.4$ | $7384.2 \pm 1650.0$ |
| 2 | $29.6 \pm 5.6$ | $28.0 \pm 5.5$ | $4.5 \pm 6.5$ | $3.5 \pm 5.5$ | $198.0 \pm 109.3$ | $195.9 \pm 107.5$ |
| 4 | $4.9 \pm 3.7$ | $4.1 \pm 3.2$ | $0.0 \pm 0.0$ | $0.0 \pm 0.0$ | $1.3 \pm 2.6$ | $1.2 \pm 2.5$ |
| 8 | $0.1 \pm 1.0$ | $0.1 \pm 0.6$ | $0.0 \pm 0.0$ | $0.0 \pm 0.0$ | $0.0 \pm 0.1$ | $0.0 \pm 0.0$ |

*Table 3.* Count statistics of $n$-grams in a canary $\hat{s}$ that also appear in the synthetic data $\widetilde{D}'$ generated using 4 reference models including and excluding $\hat{s}$. Number of $n$-grams in $\widetilde{s}$ that also appear in $\widetilde{D}'$ ($C_{unique}$), median ($C_{med}$) and average ($C_{avg}$) counts of $n$-grams from $\hat{s}$ in $\widetilde{D}'$. We report mean and std. deviation of these measures over all canaries ($F = 30$, $\mathcal{P}_{target} = 31$, $n_{rep} = 16$) for SST-2. Each canary $\hat{s}$ contains exactly 50 words and $\widetilde{D}'$ contains $685.1k \pm 45.4k$ words.

| | ROC AUC | |
|---|---|---|
| Dataset | $\epsilon = \infty$ | $\epsilon = 8$ |
| SST-2 | 0.620 | 0.48 |
| AG News | 0.654 | 0.52 |
| SNLI | 0.534 | 0.49 |

*Table 4.* ROC AUC across datasets for the strongest data-based $\mathcal{A}^{\widetilde{D}}$ MIA (2-gram), for synthetic data without ($\epsilon = \infty$) and with DP guarantees ($\epsilon = 8$). We use the setup from Table 1, i.e. synthetic canaries with natural labels, target perplexity $\mathcal{P}_{target} = 250$, no in-distribution prefix ($F = 0$) and $n_{rep} = 12$.

with access to model weights (Sablayrolles et al., 2019; Nasr et al., 2019; Leino & Fredrikson, 2020; Cretu et al., 2024), output probabilities (Shokri et al., 2017; Carlini et al., 2022a) or just labels (Choquette-Choo et al., 2021). Most powerful MIAs leverage a large number of reference models (Ye et al., 2022; Carlini et al., 2022a; Sablayrolles et al., 2019; Watson et al., 2021), while RMIA (Zarifzadeh et al., 2024) achieves high performance using only a few.

**MIAs against language models.** Song & Shmatikov (2019) study MIAs to audit the use of an individual's data during training. Carlini et al. (2021) investigate training data reconstruction attacks against LLMs, sampling synthetic text and running model-based attacks to identify likely members. Kandpal et al. (2022) and Carlini et al. (2023) both find that repetitions in the training data make records more vulnerable. Shi et al. (2024) and Meeus et al. (2024b) use attacks to identify pre-training data. Various membership scores have been proposed, e.g. model loss (Yeom et al., 2018), lowest predicted token probabilities (Shi et al., 2024), changes in the model's probability for neighboring samples (Mattern et al., 2023), or perturbations to weights (Li et al., 2023).

**Data-based MIAs in other scenarios.** Hayes et al. (2019) train a Generative Adversarial Network (GAN) on synthetic images generated by a target GAN and use the resulting discriminator to infer membership. Hilprecht et al. (2019) explore MIAs using synthetic images closest to a target record. Chen et al. (2020) study attack calibration techniques against GANs for images and location data. Privacy risks of synthetic tabular data have been widely studied, using MIAs based on similarity metrics and shadow models (Yale et al., 2019; Hyeong et al., 2022; Zhang et al., 2022). Stadler et al. (2022) compute high-level statistics, Houssiau et al. (2022) compute similarities between the target record and synthetic data, and Meeus et al. (2024a) propose a trainable feature extractor. Unlike these, we evaluate MIAs on text generated using fine-tuned LLMs. This introduces unique challenges and opportunities, both in computing membership scores and identifying worst-case canaries, making our approach distinct from prior work.

**Vulnerable records in MIAs.** Prior work found that some records (*outliers*) have a disparate effect on a trained model (Feldman & Zhang, 2020), making them more vulnerable to MIAs (Carlini et al., 2022a;b). Hence, specifically crafted canaries have been proposed to study memorization and for privacy auditing of language models, ranging from a sequence of random digits (Carlini et al., 2019; Stock et al., 2022) or tokens (Wei et al., 2024) to synthetically generated sequences (Meeus et al., 2024c). Also for synthetic tabular data, outliers have been found to have increased privacy leakage (Stadler et al., 2022; Meeus et al., 2024a).

**Decoding method.** Prior works study how decoding methods like beam search (Zanella-Béguelin et al., 2020; Carlini et al., 2023), top-$k$ sampling (Kandpal et al., 2022), or decaying temperature (Carlini et al., 2021) impact how often LLMs replicate information from their training data. We use fixed prompt templates and top-$p$ sampling with $p = 0.95$ and temperature 1.0 to assess the privacy of synthetic text in a realistic regime rather than allowing the attacker to pick a decoding method adversarially.

## Reproducibility

We provide experimental details in Section 4 and Appendix D. The datasets are publicly available, and we release the code necessary to reproduce our results on Github: https://aka.ms/canarysecho.

## Impact statement

In this work, we propose a methodology to audit the privacy risks in LLM-generated synthetic data. Through a novel MIA, we quantify the potential for sensitive information leakage even in scenarios where the underlying model is inaccessible. We also identify that canary generation mechanisms found useful to study risks in model-based attacks fall short in data-based attacks, and propose an improved canary generation mechanism optimal for data-based attacks.

Taken together, the methods proposed in this work enable an auditor to empirically estimate the privacy risks associated with synthetic text. Practitioners leveraging synthetic data as a privacy-enhancing technology can use our tools to evaluate these risks before deploying synthetic text in downstream applications. In particular, our privacy auditing pipeline would be valuable when synthetic text data is proposed to extract utility from sensitive data (e.g. medical records, financial statements) or to verify synthetic data generation implementations with formal privacy guarantees.

We hope this work advances the understanding of privacy risks in LLM-generated synthetic data and helps organizations and policymakers navigate the associated privacy-utility trade-offs effectively.

## Acknowledgements

L.W. would like to thank Robert Sim for encouraging us to work on this topic and Huseyin Inan for fruitful discussions on private synthetic data generation.

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

## A. Pseudo-code for MIAs based on synthetic data

We here provide the pseudo-code for computing membership signals for both MIA methodologies based on synthetic data (Sec. 3.1.2), see Algorithm 2 for the $n$-gram method and Algorithm 3 for the method using similarity metrics.

---

**Algorithm 2** Compute membership signal using $n$-gram model

---

1: **Parameter**: $n$-gram model order $n$
2: **Input**: Synthetic dataset $\widetilde{D} = \{\widetilde{x}_i = (\widetilde{s}_i, \widetilde{\ell}_i)\}_{i=1}^{\widetilde{N}}$, Target canary $\hat{x} = (\hat{s}, \hat{\ell})$
3: **Output**: Membership signal $\alpha$
4: $C(\vec{w}) \leftarrow 0$ for all $(n-1)$- and $n$-grams $\vec{w}$
5: **for** $i = 1$ to $\widetilde{N}$ **do**
6:      $w_1, \ldots, w_{k(i)} \leftarrow \widetilde{s}_i$
7:      **for** each $n$-gram $(w_{j-(n-1)}, \ldots, w_j)$ in $\widetilde{s}_i$ **do**
8:          $C(w_{j-(n-1)}, \ldots, w_j) \mathrel{+}= 1$
9:          $C(w_{j-(n-1)}, \ldots, w_{j-1}) \mathrel{+}= 1$
10:      **end for**
11: **end for**
12: $V \leftarrow |\{w \mid \exists i. w \in \widetilde{s}_i\}|$
13: The $n$-gram model is factored into conditional probabilities:                  {Final $n$-gram model}

$$P_{n\text{-gram}}(w_j \mid w_{j-(n-1)}, \ldots, w_{j-1}) = \frac{C(w_{j-(n-1)}, \ldots, w_j) + 1}{C(w_{j-(n-1)}, \ldots, w_{j-1}) + V}$$

14: $w_1, \ldots, w_k \leftarrow \hat{s}$                                     {Compute probability of canary text $\hat{s}$}
15: $\alpha \leftarrow \prod_{j=2}^{k} P_{n\text{-gram}}(w_j \mid w_{j-(n-1)}, \ldots, w_{j-1})$
16: **return** $\alpha$

---

---

**Algorithm 3** Compute membership signal using similarity metric

---

1: **Parameter**: Similarity metric $\text{SIM}(\cdot, \cdot)$, cutoff parameter $k$
2: **Input**: Synthetic dataset $\widetilde{D} = \{\widetilde{x}_i = (\widetilde{s}_i, \widetilde{\ell}_i)\}_{i=1}^{\widetilde{N}}$, Target canary $\hat{x} = (\hat{s}, \hat{\ell})$
3: **Output**: Membership signal $\alpha$
4: **for** $i = 1$ to $\widetilde{N}$ **do**
5:      $\sigma_i \leftarrow \text{SIM}(\hat{s}, \widetilde{s}_i)$                            {Compute similarity of each synthetic example}
6: **end for**
7: Sort similarities $\sigma_i$ for $i = 1, \ldots, \widetilde{N}$ in descending order
8: Let $\sigma_{i(1)}, \ldots, \sigma_{i(k)}$ be the top-$k$ similarities
9: $\alpha \leftarrow \frac{1}{k} \sum_{j=1}^{k} \sigma_{i(j)}$                           {Compute mean similarity of the top-$k$ examples}
10: **return** $\alpha$

---

## B. Computation of RMIA scores

We here provide more details on how we adapt RMIA, as originally proposed by Zarifzadeh et al. (2024), to our setup (see Sec. 3.1.3). In RMIA, the pairwise likelihood ratio is defined as:

$$LR_\theta(x, z) = \left(\frac{P(x \mid \theta)}{P(x)}\right) \left(\frac{P(z \mid \theta)}{P(z)}\right)^{-1}. \tag{1}$$

where $\theta$ represents the target model, $x$ the target record, and $z$ the reference population. In this work, we only consider one target model $\theta$ and many target records $x$. As we are only interested in the relative value of the likelihood ratio across target records, we can eliminate the dependency on the reference population $z$,

$$LR_\theta(x, z) = LR_\theta(x) = \frac{P(x \mid \theta)}{P(x)} \ . \tag{2}$$

As suggested by (Zarifzadeh et al., 2024), we compute $P(x)$ as the empirical mean of $P(x \mid \theta')$ across reference models $\{\theta'_i\}_{i=1}^M$,

$$P(x) = \frac{1}{M} \sum_{i=1}^M P(x \mid \theta'_i) \ . \tag{3}$$

To compute RMIA scores, we replace the probabilities in (2) by membership signals on target and reference models:

$$\beta_\theta(x) = \frac{\alpha_\theta(x)}{\frac{1}{M} \sum_{i=1}^M \alpha_{\theta'_i}(x)} \ . \tag{4}$$

Note that when we compute $\alpha_\theta(x)$ as a product of conditional probabilities (e.g. when using the target model probability in the model-based attack or the $n$-gram probability in the data-based attack), we truly use a probability for $\alpha_\theta(x)$. However, in the case of the data-based attack using similarity metrics, we use the mean similarity to the $k$ closest synthetic sequences—which does not correspond to a true probability. In this case, we normalize similarities to fall in the range $[0, 1]$ and use $\alpha_\theta(x)$ as an empirical proxy for the probability $P(x \mid \theta)$.

In practice, $P(x \mid \theta)$ can be an extremely small value, particularly when calculated as a product of token-level conditional probabilities, which can lead to underflow errors. To mitigate this, we perform arithmetic operations on log-probabilities whenever possible. However, in the context of equation (4), where the denominator involves averaging probabilities, we employ quad precision floating-point arithmetic. This method is sufficiently precise to handle probabilities for sequences of up to 50 words, which is the maximum we consider in our experiments.

## C. Prompts used to generate synthetic data

Table 5 summarizes the prompt templates $\mathsf{p}(\ell)$ used to generate synthetic data for all datasets (see Sec. 4).

| Dataset | Template $\mathsf{p}(\ell)$ | Labels $\ell$ |
|---------|------------------------------|----------------|
| SST-2   | "This is a sentence with a $\ell$ sentiment: " | {positive, negative} |
| AG News | "This is a news article about $\ell$: " | {World, Sport, Business, Sci/Tech} |
| SNLI    | "A premise with a $\ell$ hypothesis: " | {entailing, neutral, contradicting} |

*Table 5.* Prompt templates used to fine-tune models and generate synthetic data.

## D. Implementation details

To generate synthetic data throughout the experiments in this paper, we fine-tune the pre-trained model Mistral-7B (Jiang et al., 2023) using LoRA with $r = 4$, including all target modules (updating 10.7M parameters in total).

We optimized training hyperparameters for LoRA fine-tuning Mistral-7B on SST-2 by running a grid search over learning rate ($[1 \times 10^{-6}, 4 \times 10^{-6}, 2 \times 10^{-5}, 6 \times 10^{-5}, 3 \times 10^{-4}, 1 \times 10^{-3}]$) and batch size ($[64, 128, 256]$). We fine-tuned the models for 3 epochs and observed the validation loss plateaued after the first epoch. Based on these results, we selected a learning rate of $2 \times 10^{-5}$, effective batch size of 128, sequence length 128, LoRA $r = 4$ and fine-tuned the models for 1 epoch. Figure 3 shows the validation cross-entropy loss for SST-2 over the grid we searched on and the train and validation loss curves for 3 epochs with the selected hyperparameters.

All our experiments have been conducted on a cluster of nodes with 8 V100 NVIDIA GPUs with a floating point precision of 16 (`fp16`). We built our experiments on two open-source packages: (i) `privacy-estimates` which provides a

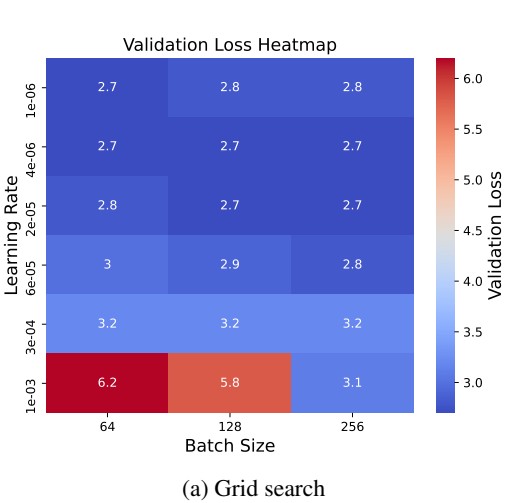

(a) Grid search

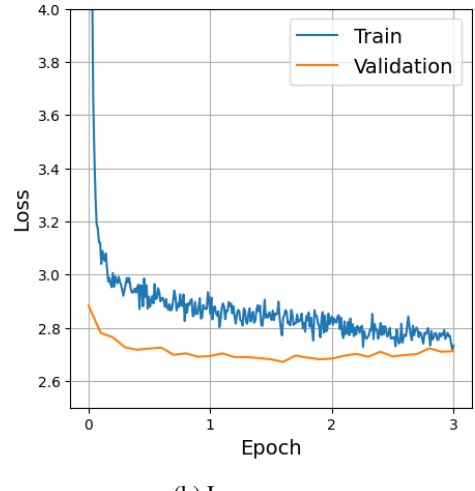

(b) Loss curve

*Figure 3.* (a) Validation cross-entropy loss of LoRA fine-tuning Mistral-7B on SST-2 varying the learning rate and effective batch size. (b) Training and validation loss for best hyperparameters over 3 epochs.

distributed implementation of the RMIA attack and (ii) `dp-transformers` which provides the implementation of the synthetic data generator.

## E. Detailed assumptions made for the adversary

We clarify the capabilities of adversaries in model- and data-based attacks according to the threat model specified in Section 2. We note:

1. A model-based attack is strictly more powerful than a data-based attack. This is because with access to the fine-tuned model $\theta$ and the prompt template $\mathsf{p}(\cdot)$, a model-based attack can synthesize $\widetilde{\mathcal{D}}$ for any set of synthetic labels and perfectly simulate the membership inference experiment for a data-based attack.

2. In both threat models, the adversary can train reference models $\{\theta_i'\}_{i=1}^M$. This assumes access to the private dataset $D$, and the training procedure of target model $\theta$, including hyperparameters. This is made clear in line 3 in Algorithm 1.

3. In our experiments, we consider model-based attacks that use the prompt template $\mathsf{p}(\cdot)$ to compute the model loss for target records, as specified in Sec. 3.1.1. Our data-based attacks use the prompt template $\mathsf{p}(\cdot)$ to generate synthetic data $\widetilde{D}$ from reference models.

4. Only the model-based attack has query-access to the target model $\theta$. The attacks used in our experiments use $\theta$ to compute token-level predicted logits for input sequences and do not use white-box features, although this is not excluded by the threat model.

5. Only the data-based attack generates synthetic data from reference models, so only this threat model leverages the sampling procedure $\mathsf{sample}(\cdot)$.

Table 6 summarizes the adversary capabilities used in the attacks in our experiments.

## F. Synthetic data utility

To ensure we audit the privacy of synthetic text data in a realistic setup, the synthetic data needs to bear high utility. We measure the synthetic data utility by comparing the downstream classification performance of RoBERTa-base (Liu et al., 2019) when fine-tuned exclusively on real or synthetic data. We fine-tune models for binary (SST-2) and multi-class classification (AG News) for 1 epoch on the same number of real or synthetic data records using a batch size of 16 and learning rate $\eta = 1 \times 10^{-5}$. We report the macro-averaged AUC score and accuracy on a held-out test dataset of real records.

| Assumptions | Model-based MIA | Data-based MIA |
|---|:---:|:---:|
| Knowledge of the private dataset $D$ used to fine-tune the target model $\theta$ (apart from knowledge of canaries). | ✓ | ✓ |
| Knowledge of the training procedure of target model $\theta$. | ✓ | ✓ |
| Knowledge of the prompt template $\mathsf{p}(\ell_i)$ used to generate the synthetic data. | ✓ | ✓ |
| Query-access to target model $\theta$, returning predicted logits. | ✓ | – |
| Access to synthetic data $\widetilde{D}$ generated by target model $\theta$. | – | ✓ |
| Knowledge of the decoding strategy employed to sample synthetic data $\widetilde{D}$ (e.g., temperature, top-$k$). | – | ✓ |

*Table 6.* Adversary capabilities effectively used by attacks in our experiments.

Table 7 summarizes the results for synthetic data generated based on original data which does not contain any canaries. While we do see a slight drop in downstream performance when considering synthetic data instead of the original data, AUC and accuracy remain high for both tasks.

| Dataset | Fine-tuning data | Classification | |
|---|---|---|---|
| | | AUC | Accuracy |
| SST-2 | Real | 0.984 | 92.3 % |
| | Synthetic | 0.968 | 91.5 % |
| AG News | Real | 0.992 | 94.4 % |
| | Synthetic | 0.978 | 90.0 % |

*Table 7.* Utility of synthetic data generated from real data *without* canaries. We compare the performance of text classifiers trained on real or synthetic data—both evaluated on real, held-out test data.

We further measure the synthetic data utility when the original data contains standard canaries (see Sec. 5.1). Specifically, we consider synthetic data generated from a target model trained on data containing 500 canaries repeated $n_{\text{rep}} = 12$ times, so 6000 data records. When inserting canaries with an artificial label, we remove all synthetic data associated with labels not present originally when fine-tuning the RoBERTa-base model.

| Dataset | Canary injection | | Classification | |
|---|---|---|---|---|
| | Source | Label | AUC | Accuracy |
| SST-2 | In-distribution | | 0.972 | 91.6 % |
| | Synthetic | Natural | 0.959 | 89.3 % |
| | | Artificial | 0.962 | 89.9 % |
| AG News | In-distribution | | 0.978 | 89.8 % |
| | Synthetic | Natural | 0.977 | 88.6 % |
| | | Artificial | 0.980 | 90.1 % |

*Table 8.* Utility of synthetic data generated from real data *with* canaries ($n_{\text{rep}} = 12$). We compare the performance of text classifiers trained on real or synthetic data—both evaluated on real, held-out test data.

Table 8 summarizes the results. Across all canary injection methods, we find limited impact of canaries on the downstream utility of synthetic data. While the difference is minor, the natural canary labels lead to the largest utility degradation. This makes sense, as the high perplexity synthetic sequences likely distort the distribution of synthetic text associated with a certain real label. In contrast, in-distribution canaries can be seen as up-sampling certain real data points during fine-tuning, while canaries with artificial labels merely reduce the capacity of the model to learn from real data and do not interfere with this process as much as canaries with natural labels do.

## G. Additional results for MIAs using standard canaries

In line with the literature on MIAs against machine learning models (Carlini et al., 2022a), we also evaluate MIAs by their true positive rate (FPR) at low false positive rates (FPR). Tables 9 and 10 summarize the MIA TPR at FPR=0.01 and FPR=0.1, respectively. We also provide the ROC curves for the data-based MIAs for both datasets, considering canaries with natural labels in Figure 4.

| Dataset | Canary injection | | TPR@FPR=0.01 | | | |
| | Source | Label | Model $\mathcal{A}^\theta$ | Synthetic $\mathcal{A}^{\widetilde{D}}$ (2-gram) | Synthetic $\mathcal{A}^{\widetilde{D}}$ (SIM$_{\text{Jac}}$) | Synthetic $\mathcal{A}^{\widetilde{D}}$ (SIM$_{\text{emb}}$) |
|---|---|---|---|---|---|---|
| SST-2 | In-distribution | | 0.148 | 0.104 | 0.029 | 0.020 |
| | Synthetic | Natural | 0.972 | 0.042 | 0.018 | 0.024 |
| | | Artificial | 0.968 | 0.057 | 0.000 | 0.030 |
| AG News | In-distribution | | 0.941 | 0.050 | 0.032 | 0.016 |
| | Synthetic | Natural | 0.955 | 0.049 | 0.006 | 0.016 |
| | | Artificial | 0.990 | 0.053 | 0.041 | 0.022 |

*Table 9.* True positive rate (TPR) at a false positive rate (FPR) of 0.01 for experiments using standard canaries (Sec. 5.1) across training datasets, threat models (model-based adversary $\mathcal{A}^\theta$ and data-based adversary $\mathcal{A}^{\widetilde{D}}$) and MIA methodologies. Canaries are synthetically generated with target perplexity $\mathcal{P}_{\text{target}} = 250$, with no in-distribution prefix ($F = 0$) and inserted $n_{\text{rep}} = 12$ times.

| Dataset | Canary injection | | TPR@FPR=0.1 | | | |
| | Source | Label | Model $\mathcal{A}^\theta$ | Synthetic $\mathcal{A}^{\widetilde{D}}$ (2-gram) | Synthetic $\mathcal{A}^{\widetilde{D}}$ (SIM$_{\text{Jac}}$) | Synthetic $\mathcal{A}^{\widetilde{D}}$ (SIM$_{\text{emb}}$) |
|---|---|---|---|---|---|---|
| SST-2 | In-distribution | | 0.795 | 0.406 | 0.207 | 0.203 |
| | Synthetic | Natural | 0.996 | 0.191 | 0.114 | 0.128 |
| | | Artificial | 1.000 | 0.277 | 0.142 | 0.142 |
| AG News | In-distribution | | 0.982 | 0.314 | 0.158 | 0.168 |
| | Synthetic | Natural | 0.990 | 0.271 | 0.114 | 0.114 |
| | | Artificial | 0.996 | 0.323 | 0.152 | 0.164 |

*Table 10.* True positive rate (TPR) at a false positive rate (FPR) of 0.1 for experiments using standard canaries (Sec. 5.1) across training datasets, threat models (model-based adversary $\mathcal{A}^\theta$ and data-based adversary $\mathcal{A}^{\widetilde{D}}$) and MIA methodologies. Canaries are synthetically generated with target perplexity $\mathcal{P}_{\text{target}} = 250$, with no in-distribution prefix ($F = 0$) and inserted $n_{\text{rep}} = 12$ times.

## H. Ablations for MIAs on synthetic data

**Synthetic multiple**    Thus far, we have exclusively considered that the number of generated synthetic records equals the number of records in the real data, i.e., $N = \widetilde{N}$. We now consider the case when more synthetic data is made available to a data-based adversary ($\widetilde{\mathcal{A}}$). Specifically, we denote the *synthetic multiple* $m = \widetilde{N}/N$ and evaluate how different MIAs perform for varying values of $m$. Figure 5 shows how the ROC AUC score varies as $m$ increases. As expected, the ROC AUC score for the attack that uses membership signals computed using a 2-gram model trained on synthetic data increases when more synthetic data is available. In contrast, attacks based on similarity metrics do not seem to benefit significantly from this additional synthetic data.

**Hyperparameters in data-based attacks**    The data-based attacks that we presented in Sec. 3.1 rely on certain hyperparameters. The attack that uses $n$-gram models to compute membership signals is parameterized by the order $n$. Using a too small value for $n$ might not suffice to capture the information leaked from canaries into the synthetic data used to train the $n$-gram model. When using a too large order $n$, on the other hand, we would expect less overlap between $n$-grams present in the synthetic data and the canaries, lowering the membership signal.

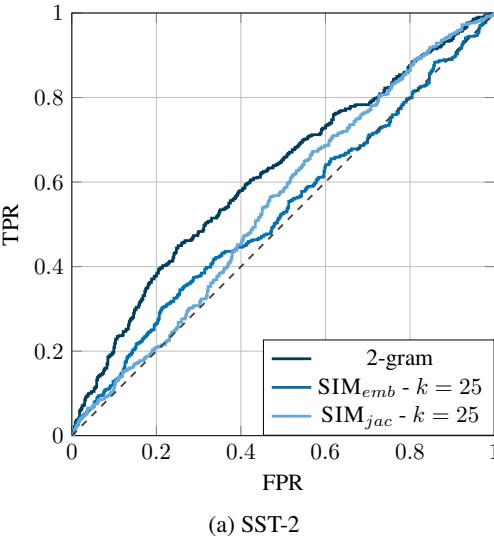
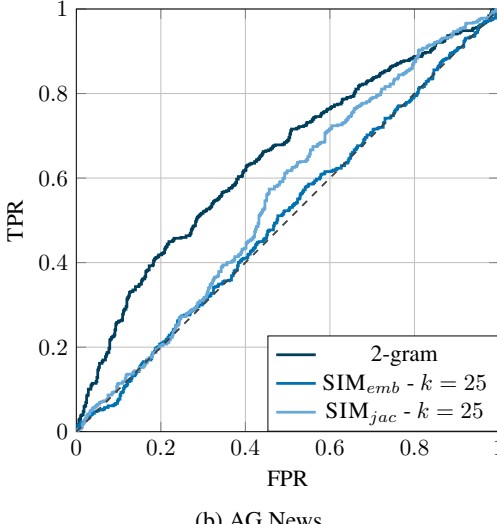

(a) SST-2          (b) AG News

*Figure 4.* MIA ROC curves across data-based MIA methodologies for the SST-2 (left) and AG News (right) datasets. Canaries are synthetically generated with target perplexity of $\mathcal{P}_{\text{target}} = 250$ with a natural label, with no in-distribution prefix ($F = 0$) and inserted $n_{\text{rep}} = 12$ times.

Further, the similarity-based methods rely on the computation of the mean similarity of the closest $k$ synthetic records to the a canary. When $k$ is very small, e.g. $k = 1$, the method takes into account a single synthetic record, potentially missing on leakage of membership information from other close synthetic data records. When $k$ becomes too large, larger regions of the synthetic data are taken into account, which might dilute the membership signal among the noise.

Table 11 reports the ROC AUC scores of data-based attacks for different values of the hyperparameters $n$ and $k$ when using standard canaries (Sec. 5.1). We find that for both datasets, training a 2-gram model on the synthetic data to compute the membership signal yields the best performance. For the data-based MIAs relying on the similarity between the canary and the synthetic records, both when considering Jaccard distance and cosine distance in the embedding space, we find that considering the $k = 25$ closest synthetic records yields the best performance.

| Dataset | $n$-gram | | SIM$_{\text{Jac}}$ | | SIM$_{\text{emb}}$ | |
| | $n$ | AUC | $k$ | AUC | $k$ | AUC |
|---|---|---|---|---|---|---|
| SST-2 | 1 | 0.415 | 1 | 0.520 | 1 | 0.516 |
| | 2 | **0.616** | 5 | 0.535 | 5 | 0.516 |
| | 3 | 0.581 | 10 | 0.538 | 10 | 0.519 |
| | 4 | 0.530 | 25 | **0.547** | 25 | **0.530** |
| AG News | 1 | 0.603 | 1 | 0.522 | 1 | 0.503 |
| | 2 | **0.644** | 5 | 0.525 | 5 | 0.498 |
| | 3 | 0.567 | 10 | 0.537 | 10 | 0.503 |
| | 4 | 0.527 | 25 | **0.552** | 25 | **0.506** |

*Table 11.* Ablation over hyperparameters of data-based MIAs. We report ROC AUC scores across different values of the hyperparameters $n$ and $k$ (see Sec. 3.1). Canaries are synthetically generated with target perplexity $\mathcal{P}_{\text{target}} = 250$, with a natural label, with no in-distribution prefix ($F = 0$), and inserted $n_{\text{rep}} = 12$ times.

## I. Disparate vulnerability of standard canaries

We analyze the disparate vulnerability of standard canaries between the model-based attack and the data-based attack that uses a 2-gram model (as discussed in Sec 5.1). Figure 6 plots the RMIA scores for both attacks on the same set of canaries, which have either been included in the training dataset of the target model (*member*) or not (*non-member*). Note that the RMIA scores are used to distinguish members from non-members, and that a larger value corresponds to the adversary

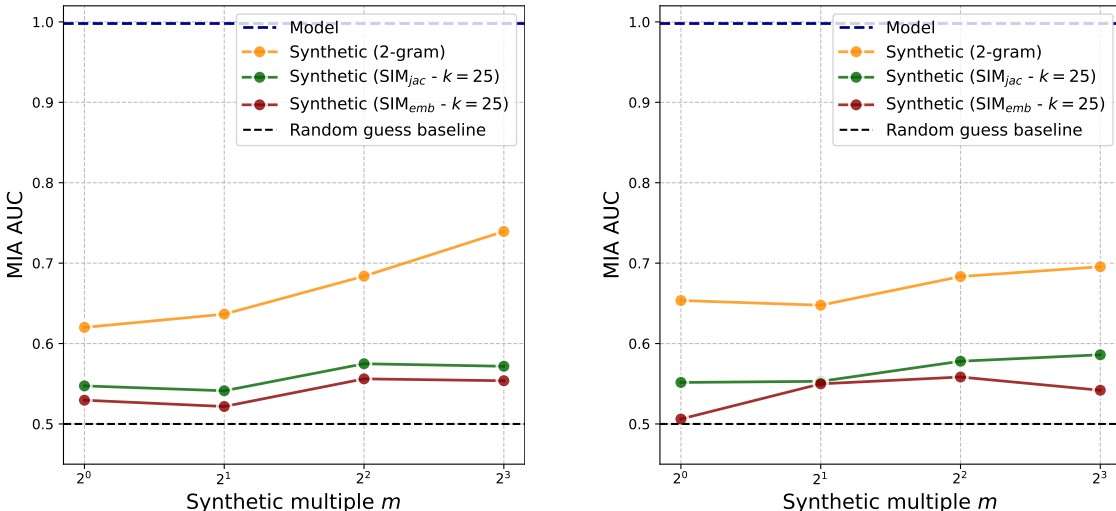

*Figure 5.* ROC AUC score for increasing value of the synthetic multiple $m$ across data-based attack methods for SST-2 (left) and AG News (right). Canaries are synthetically generated with target perplexity of $\mathcal{P}_{\text{target}} = 250$, with a natural label, with no in-distribution prefix ($F = 0$), and inserted $n_{\text{rep}} = 12$ times.

being more confident in identifying a record as a member, i.e., to the record being more *vulnerable*.

First, we note that the scores across both threat models exhibit a statistically significant, positive correlation. We find a Pearson correlation coefficient between the RMIA scores (log) for both methods of $0.20$ ($p$-value of $2.4 \times 10^{-10}$) and $0.23$ ($p$-value of $1.9 \times 10^{-13}$) for SST-2 and AG News, respectively. This means that a record vulnerable to the model-based attack tends to be also vulnerable to the data-based attack, even though the attacks differ substantially.

Second, and more interestingly, some canaries have disparate vulnerability across MIA methods. Indeed, Figure 6 shows how certain data records which are not particularly vulnerable to the model-based attack are significantly more vulnerable to the data-based attack, and vice versa.

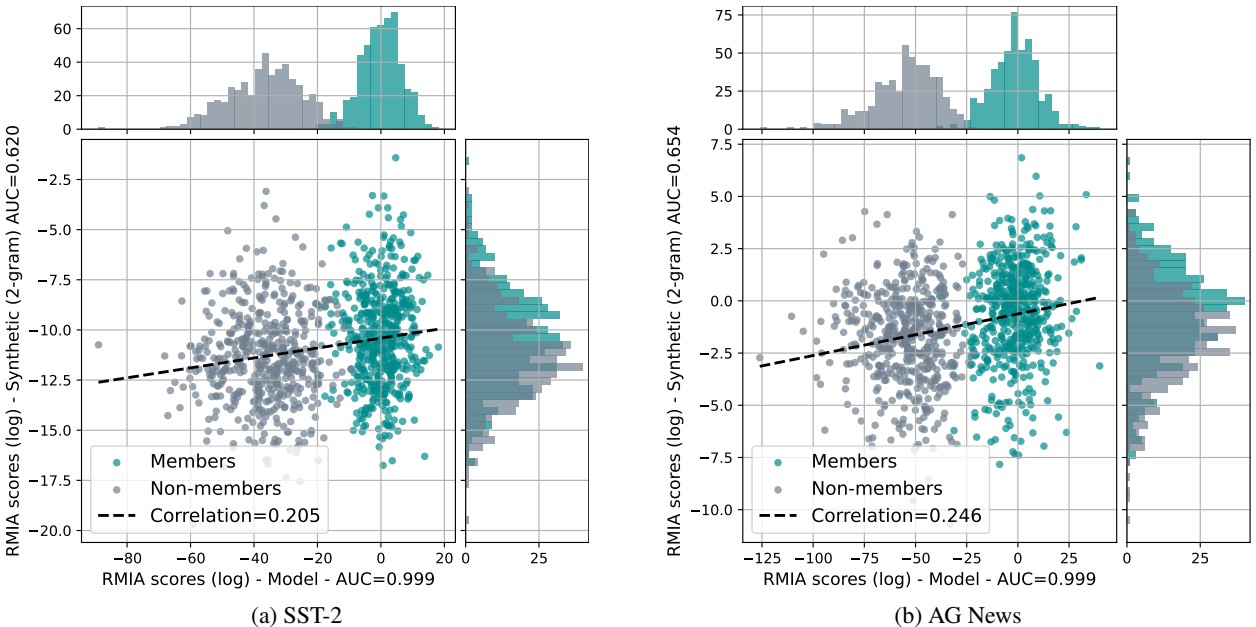

*Figure 6.* RMIA scores (log) for model- and data-based MIAs on the same set of canaries. Results for both datasets SST-2 and AG News. Canaries are synthetically generated with target perplexity of $\mathcal{P}_{\text{target}} = 250$ with a natural label, and inserted $n_{\text{rep}} = 12$ times.

# J. Low FPR ROC results

Figure 7 shows log-log plots of the ROC curves in Figure 1 to better examine behavior of attacks at low FPR.

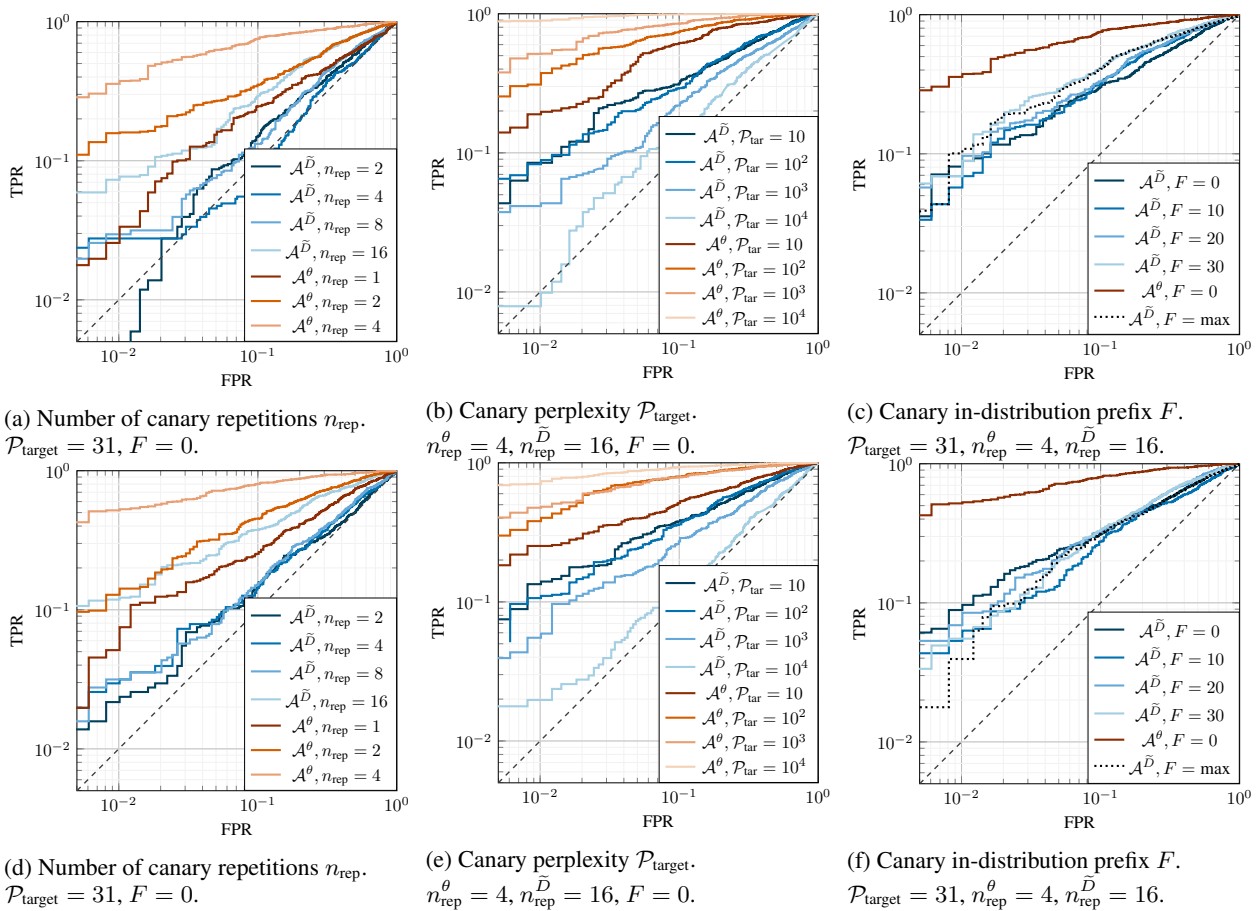

(a) Number of canary repetitions $n_{\text{rep}}$. $\mathcal{P}_{\text{target}} = 31$, $F = 0$.

(b) Canary perplexity $\mathcal{P}_{\text{target}}$. $n_{\text{rep}}^{\theta} = 4$, $n_{\text{rep}}^{\widetilde{D}} = 16$, $F = 0$.

(c) Canary in-distribution prefix $F$. $\mathcal{P}_{\text{target}} = 31$, $n_{\text{rep}}^{\theta} = 4$, $n_{\text{rep}}^{\widetilde{D}} = 16$.

(d) Number of canary repetitions $n_{\text{rep}}$. $\mathcal{P}_{\text{target}} = 31$, $F = 0$.

(e) Canary perplexity $\mathcal{P}_{\text{target}}$. $n_{\text{rep}}^{\theta} = 4$, $n_{\text{rep}}^{\widetilde{D}} = 16$, $F = 0$.

(f) Canary in-distribution prefix $F$. $\mathcal{P}_{\text{target}} = 31$, $n_{\text{rep}}^{\theta} = 4$, $n_{\text{rep}}^{\widetilde{D}} = 16$.

*Figure 7.* Log-log ROC curves of MIAs on synthetic data $\mathcal{A}^{\widetilde{D}}$ compared to model-based MIAs $\mathcal{A}^{\theta}$ on SST-2 (7a–7c) and AG News (7d–7f). We ablate over the number of canary insertions $n_{\text{rep}}$ in 7a, 7d, the target perplexity $\mathcal{P}_{\text{target}}$ of the inserted canaries in 7b, 7e and the length $F$ of the in-distribution prefix in the canary in 7c, 7f.

# K. Interpretability

To further understand the membership signal for data-based attacks, we examine some examples in-depth.

Specifically, we consider the MIA for specialized canaries with $F = 30$, $\mathcal{P}_{\text{target}} = 31$ and $n_{\text{rep}} = 16$ for SST-2 from Figure 1c. Recall that for this attack, we consider 1000 canaries, 500 of which are injected into the training dataset of one target model $\theta$. We also train 4 references models $\{\theta'_i\}_{i=1}^{4}$ where each of the 1000 canaries has been included in exactly half. We focus on the best performing MIA based on synthetic data, i.e. the attack leveraging the probability of the target sequence computed using a 2-gram model trained on the synthetic data.

To understand what signal the MIA picks up to infer membership, we focus on the canary most confidently, and correctly, identified as member and the one most confidently, and correctly, identified as non-member. For this, we take the canaries for which the RMIA score computed using the target model and the reference models is the highest and the lowest, respectively.

Next, for each model (4 reference models, and 1 target model), we report for this canary $\hat{x}_i$:

1. Whether the canary has been included in, $\hat{x}_i \in D$ (IN), or excluded from, $\hat{x}_i \notin D$ (OUT), the training dataset of the model in question, and thus to generate the synthetic data $\widetilde{D} = \{\widetilde{x}_i = (\widetilde{s}_i, \widetilde{\ell}_i)\}_{i=1}^{\widetilde{N}}$.

2. The canary with the words that appear as a 2-gram in the synthetic data $\widetilde{D}$ emphasized in bold face. Note that if, for instance, this is a sequence of 3 words, e.g., *"the woodman seems"*, this means that all 3 words appear in 2-grams in the synthetic data, e.g., *"the woodman"* and *"woodman seems"*.

3. The maximum overlapping sub-string between the canary and any synthetically generated record $\widetilde{s}_i$. We define a sub-string as a sequence of characters, including white space, and also report its length as number of characters $L_{\text{overlap}}$.

4. The mean, negative cross-entropy loss of the canary computed using the 2-gram model trained on the synthetic data. Formally, for canary $\hat{s}_i = (w_1, w_2, \ldots, w_k)$: $-\frac{1}{k} \sum_{j=2}^{k} \log \left( P_{\text{2-gram}}(w_j, w_{j-1}) \right)$.

Tables 12 and 13 report this for the canary with the largest and lowest RMIA score, respectively.

First, we analyze the membership prediction made for the canary with the largest RMIA score (Table 12). Examining the reference models ($\theta'_i$), we find little variation in the metrics we consider, regardless of whether the canary was included in the training dataset (IN) or not (OUT). Specifically, the number of overlapping 2-grams, the length of the longest overlapping sub-string, and the 2-gram loss remain largely unchanged across IN and OUT reference models.

In contrast, the target model $\theta$ exhibits a strong signal, especially when compared to the reference models. Notably, the uncommon sequence *"Embed from Getty Images Embed from Getty Images"* appears in the synthetic data generated by the trained target model $\theta$ but is absent from the synthetic data of all $\theta'_i$. The signal is further reflected by a significantly lower 2-gram loss compared to the reference models, explaining the high RMIA score for this canary.

Overall, even for the most vulnerable canary, not all of its 2-grams appear in the synthetic data, and the longest overlapping sub-string accounts for only 52 out of 296 characters. This suggests that membership inference does not rely on verbatim regurgitation of long sub-sequences. Instead, it detects subtler patterns, such as the presence of specific 2-grams or shorter sub-strings. Such signal is effectively captured by the 2-gram loss and becomes especially meaningful when contrasted against values reached for the reference models using RMIA.

Second, we analyze the membership prediction for the canary with the lowest RMIA score (Table 13). In this case, the canary was not included in the target model's training dataset (OUT) and was correctly classified as non-member.

We observe minimal differences in the number of overlapping 2-grams and the length of the longest overlapping sub-string across IN and OUT reference models, as well as the target model. Instead, the most informative signal emerges from the 2-gram loss: it is lower for IN models than for OUT models, with the target model exhibiting the highest loss, resulting in the low RMIA score. These results again suggest that the information useful to infer membership based on synthetic data does not rely on the regurgitation of long sub-sequences, and instead arises from slight shifts in the probability distribution of co-occurrences of words in the synthetic data, as captured by the 2-gram loss.

| Model | IN or OUT | Canary (words present as part of 2-grams in $\widetilde{D}'$ in bold) | Max overlapping sub-string | 2-gram loss |
|---|---|---|---|---|
| $\theta'_1$ (ref) | IN | "**the woodman seems to have** directly influenced this girl-meets-girl **love story , but even more** reassuring **is how its makers actually seem to understand what made allen 's romantic comedies work in the first place.** Embed from Getty Images Embed from Getty Images Earlier this week, **the case against the**" | « *to understand what made* » ; $L_{\text{overlap}} = 25$ | 8.21 |
| $\theta'_2$ (ref) | IN | "the woodman **seems to have** directly influenced this girl-meets-girl **love story , but even more** reassuring **is how its makers** actually **seem to understand what made allen 's romantic comedies work in the first place.** Embed from Getty Images Embed from Getty Images Earlier this **week, the case against the**" | «*ally seem to understand* » ; $L_{\text{overlap}} = 24$ | 8.19 |
| $\theta'_3$ (ref) | OUT | "the woodman **seems to have** directly influenced this girl-meets-girl **love story , but even more** reassuring **is how its makers** actually **seem to understand what made allen 's romantic comedies work in the first place.** Embed from Getty Images Embed from Getty Images Earlier **this week, the case against the**" | « *seem to understand what ma*» ; $L_{\text{overlap}} = 27$ | 8.18 |
| $\theta'_4$ (ref) | OUT | "the woodman **seems to have** directly influenced this girl-meets-girl **love story , but even more** reassuring **is how its makers** actually **seem to understand what made allen 's romantic comedies work in the first place.** Embed from Getty Images Embed from Getty Images Earlier this week, **the case against the**" | «*s work in the first place*» ; $L_{\text{overlap}} = 25$ | 8.18 |
| $\theta$ (target) | IN | "the woodman **seems to have** directly influenced this girl-meets-girl **love story , but even more** reassuring **is how its makers actually seem to understand what made allen 's romantic comedies work in the first place. Embed from Getty Images Embed from Getty Images** Earlier this week, **the case against the**" | «*e. Embed from Getty Images Embed from Getty Images E*» ; $L_{\text{overlap}} = 52$ | 7.59 |

*Table 12.* Interpretability of the best MIA (2-gram) based on synthetic data for specialized canaries with $F = 30$, $\mathcal{P}_{\text{target}} = 31$ and $n_{\text{rep}} = 16$ for SST-2 from Figure 1c. Results across 4 reference models and the target model for the canary with the **largest RMIA score** (most confidently and correctly identified as member by the MIA). Words in bold appear in 2-grams in $\widetilde{D}'$. The largest generated sub-sequence of the canary in $\widetilde{D}'$ corresponds to the maximum overlapping sub-string, not the longest sequence of words in bold.

| Model | IN or OUT | Canary (words present as part of 2-grams in $\widetilde{D}'$ in bold) | Max overlapping sub-string | 2-gram loss |
|---|---|---|---|---|
| $\theta'_1$ (ref) | IN | **"give a spark to " chasing amy " and " changing lanes " falls flat as thinking man** cia agent **jack ryan in this summer 's new action film , " the sum of all fears , "** in theaters friday . **if director philip** noyce **and writer** aaron singer" | « " *the sum of all fears , " » ;* $L_{\text{overlap}} = 30$ | 7.80 |
| $\theta'_2$ (ref) | IN | **"give a spark to " chasing amy " and " changing lanes " falls flat as thinking man cia agent jack ryan in this summer 's new action film , " the sum of all fears , " in theaters friday . if director philip noyce and writer** aaron singer" | «, " *the sum of all fears ', » ;* $L_{\text{overlap}} = 26$ | 7.73 |
| $\theta'_3$ (ref) | OUT | **"give a spark to "** chasing amy **" and "** changing lanes **" falls flat as thinking man** cia agent jack ryan **in this summer 's new action film , " the sum of all fears , "** in theaters friday . **if director philip noyce and writer** aaron singer" | « , " *the sum of all fears » ;* $L_{\text{overlap}} = 27$ | 8.27 |
| $\theta'_4$ (ref) | OUT | **"give a spark to " chasing amy " and " changing lanes " falls flat as thinking man** cia agent jack ryan **in this summer 's new action film , " the sum of all fears , "** in theaters friday . **if director philip noyce and writer** aaron singer" | « " *chasing amy " and " changing lanes » ;* $L_{\text{overlap}} = 41$ | 7.99 |
| $\theta$ (target) | OUT | **"give a spark to " chasing amy " and "** changing lanes **" falls flat as thinking man** cia agent jack ryan **in this summer 's new action film , " the sum of all fears , "** in theaters friday . **if director** philip noyce **and writer** aaron singer" | « " *the sum of all fears , " » ;* $L_{\text{overlap}} = 30$ | 8.30 |

*Table 13.* Interpretability of the best MIA (2-gram) based on synthetic data for specialized canaries with $F = 30$, $\mathcal{P}_{\text{target}} = 31$ and $n_{\text{rep}} = 16$ for SST-2 from Figure 1c. Results across 4 reference models and the target model for the canary with the **smallest RMIA score** (most confidently and correctly identified as non-member by the MIA). Words in bold appear in 2-grams in $\widetilde{D}'$. The largest generated sub-sequence of the canary in $\widetilde{D}'$ corresponds to the maximum overlapping sub-string, not the longest sequence of words in bold.
.

