# OpenReview forum: "The Canary’s Echo: Auditing Privacy Risks of LLM-Generated Synthetic Text"
_ICML.cc/2025/Conference — ICML 2025 poster_

### Official Review · Reviewer_Ni1p · 2025-03-01

**Overall Recommendation:** 3

**Summary:**

This paper argues that the synthetic data generated by LLMs, which are finetuned on private data, poses privacy risks. They identify a new class of canaries suited for these data-based privacy risks, and show that by choosing an in-distribution prefix and out-of-distribution suffix they can greatly increase the vulnerability of the synthetic data.

## update after rebuttal
After the rebuttal, in which I mainly asked the authors to add some additional baselines/ablations, I keep my score.

**Claims And Evidence:**

I think all the claims in this submission are well-substantiated.  There are extensive ablations throughout the entire manuscript.

**Essential References Not Discussed:**

N/A

**Experimental Designs Or Analyses:**

Yes. I checked the main MIA experimental design.

**Methods And Evaluation Criteria:**

Yes, the metrics reported are standard in the MIA literature.

**Other Comments Or Suggestions:**

It might be worthwhile to include a discussion on the privacy risks of $\mathcal{D}$ itself when it does not have any canaries and just given access to the synthetic data itself. I imagine that the performance of the MIAs would be much worse, since the canaries are specially designed to be memorized.

**Other Strengths And Weaknesses:**

Strengths:
1. The paper is very well written.
2. The framing of the data-based attacks for the privacy risks of synthetic data is important for practice.
3. The new construction of canaries with in-distribution prefixes and out of distribution suffixes is very original and interesting, and the authors do a good of ablating and thoroughly exploring this design space.

Weaknesses:
1. The authors do not fully explore the design space of MIAs for synthetic dataset. For example, one could imagine finetuning on $\tilde{\mathcal{D}}$ and then applying standard MIAs to the finetuned model to try to extract canaries from $\mathcal{D}$. This is not a major concern because the MIAs solely based on synthetic data are already performant.

**Questions For Authors:**

1. Could you report CIs for some of the major results, i.e. Table 1 and Figure 1?
2. The text in Figure 3a is quite small; could you the plot more legible?
3. Could you explore the effect of training data size on the success of the MIAs? My intuition is that if the training data size goes to infinity, then one needs more repetitions of the canary.
4. Could you report the performance of the data-based MIAs on non-canaries belonging to $\mathcal{D}$ as a baseline? Are there privacy risks inherent in any finetuning data given synthetic generations from the finetuned model?

**Relation To Broader Scientific Literature:**

Existing literature has found that machine learning models trained on private data pose substantial privacy risks as quantified by MIAs (Shokri et al. 2017, Carlini et al. 2022a, Shi et al. 2023). However, the privacy risks of synthetic data generated by LLMs has not been explored by these works. This submission initiates a thorough investigation into these new risks and show that one can construct canaries which can be identified in the synthetic data by simple MIAs.

**Theoretical Claims:**

N/A

---

> ### Author Rebuttal · Authors · 2025-04-01
>
> We thank the reviewer for their feedback. We provide responses for the concerns raised below.
>
> > (1) one could imagine finetuning on D_tilde and then applying standard MIAs to the finetuned model
>
> Many thanks for pointing this out. We have opted to train an n-gram model on the synthetic data rather than a larger, transformer-based model due to its simplicity and computational cost. Indeed, training the n-gram model on the synthetic data, both for SST-2 and AgNews takes less than 1 CPU minute. We will add the suggestion of training a full LLM on the synthetic data instead to the discussion section.
>
> > (2) include a discussion on the privacy risks of D
>
> We provide results on fully in-distribution canaries (randomly sampled from D, no out-of-distribution suffix or F=max) throughout our work (in-distribution results in Table 1, and results for F=max in Figure 2 (c,f) and Table 2). Due to the lower perplexity of in-distribution sequences, we recover that data-based MIAs work quite well for these samples, especially compared to the high-perplexity canaries commonly used for model-based attacks. However, in these experiments, we consider the member canaries repeated n_rep times in the training data with n_rep up to 12 and 16. From Figure 2(a,d), we learn that when n_rep decreases, the MIA performance drops to no better than a random guess baseline. We hence anticipate that, at least in our experimental setup, the privacy risks associated with sequences appearing only once in D remains low. We will elaborate on this in the discussion section. We believe this also answers the reviewer’s last question (i.e. reporting the performance of the data-based MIAs on non-canaries belonging to D as a baseline).
>
> > (3) Could you report CIs for some of the major results
>
> For our main results (e.g. Table 1), we report ROC AUC as the performance of the MIA, representing an average performance over all (1000) canaries.
>
> Getting meaningful confidence intervals for these results requires training multiple (10+) target models, which is computationally quite expensive and was not feasible within the rebuttal period. We will run this for the SST-2 results in Table 1 to be included in a final version of the paper.
>
> > (4) The text in Figure 3a is quite small; could you make the plot more legible?
>
> Thanks for pointing this out, we will increase the corresponding fontsize.
>
> > (5) Could you explore the effect of training data size on the success of the MIAs?
>
> We share the reviewer’s intuition that as the size of the training dataset increases, the MIA performance likely decreases. As part of the rebuttal process, however, we have prioritized running other experiments and would leave this analysis to future work.

---

> > ### Comment · Reviewer_Ni1p · 2025-04-01
> >
> > Thank you for your detailed rebuttal. I keep my score.

---

> > > ### Author Response · Authors · 2025-04-09
> > >
> > > We promised earlier (rebuttal for reviews above) to run 2 additional experiments.
> > >
> > > **1. MIA results for synthetic data with formal privacy guarantees** (L9fq, qpCC, MC8u)
> > >
> > > We hypothesized that MIAs against synthetic data generated from models fine-tuned with DP guarantees would approach random guess performance (AUC of 0.5). We run additional experiments to determine whether this intuition is correct.
> > >
> > > Below, we provide the MIA AUC for the best data-based attack (2-gram) when the target model is fine-tuned with DP-SGD with ε=8, under the setup of Table 1 in the paper (column *Synthetic 𝒜^𝐷 (2-gram)* vs *Synthetic 𝒜^𝐷_DP (2-gram)*). New results are in bold.
> > >
> > > We confirm the MIA AUC to be close to 0.5, providing strong evidence that DP constitutes a strong defense against data-based MIAs. We also find that the corresponding generated synthetic data maintains a high utility in downstream tasks. Specifically, for synthetic data generated with ε=8, accuracy on SST-2 reaches 91.6%, compared to 91.5% for non-DP synthetic data and 92.3% for real data (Table 6).
> > >
> > > | Dataset   | Source  | Label | Model 𝒜^θ | Synthetic 𝒜^𝐷 (2-gram) | Synthetic 𝒜^𝐷_DP (2-gram) | Synthetic 𝒜^𝐷 (SIM_Jac) | Synthetic 𝒜^𝐷 (SIM_emb) |
> > > |-|-|-|-|-|-|-|-|
> > > | SST-2     | In-distribution |   | 0.911    | 0.741| **0.49**| 0.602  | 0.586|
> > > | | Synthetic   | Natural  |  0.999  | 0.620| **0.48** | 0.547| 0.530 |
> > > |  |  | Artificial  | 0.999  |0.682  | **0.50** | 0.552 | 0.539 |
> > > | AG News   | In-distribution |     |   0.993   | 0.676 | **0.52** | 0.590  | 0.565 |
> > > |  | Synthetic   | Natural    | 0.996  |0.654| **0.52** | 0.552  | 0.506  |
> > > |  |    | Artificial  | 0.999  |0.672 |**0.51**  | 0.560  | 0.525 |
> > > | **SNLI**  | In-distribution |       | **0.892**  | **0.718**  | **0.511**  | **0.644** | **0.630**|
> > > |   | Synthetic   | Natural     | **0.998**  | **0.534**  | **0.49**  | **0.486**  | **0.488**|
> > > |  |  | Artificial  | **0.997**  |**0.770**  | TBD  | **0.602**  | **0.571** |
> > >
> > > Reviewer MC8u suggests readers could benefit from a discussion on defenses against MIAs, specifically on methods that offer DP guarantees. We concur and provide a discussion below that we will incorporate into the paper to complement the survey of methods to synthesize text with DP guarantees in Section 2 in the submission and the results we share above on synthetic data generated from models fine-tuned with DP-SGD.
> > >
> > > **Discussion on defenses.** Methods to generate synthetic text with DP guarantees mitigate MIAs by ensuring that any single training record exerts limited influence on synthesized data. These methods are broadly split into training-time [A,B,C] and inference-time [D,E,F,G]. We focus on the former, specifically on methods that fine-tune a pre-trained LLM with DP-SGD and then prompt this model to generate synthetic data. Training-time methods leverage the post-processing property of DP to transfer the guarantees from the fine-tuned model to synthetic data. Because generating synthetic data from a DP model does not consume additional privacy budget, they can generate an unlimited amount of data with a fixed privacy budget. In contrast, inference-time methods use unmodified pre-trained models prompted on private data and inject calibrated noise during decoding [E,F,G] or employ DP evolutionary algorithms to steer generation towards a distribution similar to the private data [D].
> > >
> > > Empirical evaluation suggests that DP synthetic text can achieve high utility. Our results provide additional evidence of this and also that DP constitutes a strong mitigation against data-based MIAs. As the field progresses, we expect that rigorous privacy auditing using MIAs adapted to actual threat models will be crucial to the adoption of synthetic text generation.
> > >
> > > [A] Yue et al., Synthetic text generation with differential privacy: A simple and practical recipe. ACL 2023
> > >
> > > [B] Mattern et al., Differentially Private Language Models for Secure Data Sharing. EMNLP 2022
> > >
> > > [C] Kurakin et al., Harnessing large-language models to generate private synthetic text. ICLR 2024
> > >
> > > [D] Xie et al., Differentially private synthetic data via foundation model APIs 2: Text. ICML 2024
> > >
> > > [E] Wu et al., Privacy-Preserving In-Context Learning for Large Language Models. ICLR 2024
> > >
> > > [F] Tang et al., Privacy-Preserving In-Context Learning with Differentially Private Few-Shot Generation. ICLR 2024
> > >
> > > [G]  Amin et al., Private prediction for large-scale synthetic text generation. EMNLP 2024
> > >
> > > **2. Results for a third dataset** (L9fq, MC8u)
> > >
> > > We also conducted experiments for a third dataset in the Table above. Specifically, we consider the SNLI dataset, and report the MIA AUC for the model-based attack and all three data-based attacks. We confirm that the data-based attacks also work for this dataset and recover an MIA performance drop compared to model-based MIAs similar to the one observed for the other two datasets. We will propagate other results for SNLI in an eventual final version of the paper.

---

### Official Review · Reviewer_MC8u · 2025-03-08

**Overall Recommendation:** 4

**Summary:**

The paper investigates the privacy risks associated with releasing synthetic data generated by Large Language Models (LLMs). It explores how much information about the original training data can be extracted from such synthetic data, even when adversaries do not have direct access to the fine-tuned model.


- **Synthetic Data Leakage:** MIAs using only synthetic data can detect membership with AUC scores significantly above random, showing that synthetic text leaks training information.
- **Attack Comparison:** There's a gap between model-based and data-based attacks; canaries effective in one setting require much higher occurrence to be vulnerable in the synthetic data scenario.
- **Improved Canary Design:** The paper proposes canaries with an in-distribution prefix and high-perplexity suffix, enhancing their detectability in synthetic outputs for more reliable privacy auditing.

**Claims And Evidence:**

Yes

**Essential References Not Discussed:**

NA

**Experimental Designs Or Analyses:**

Yes.

**Methods And Evaluation Criteria:**

Yes

**Other Comments Or Suggestions:**

How do attacks work on differentially private text generation (text-to-text privatization) [1, 2, 3, 4]? Recent studies [2, 4] have demonstrated that paraphrasing techniques can achieve a highly favorable privacy-utility trade-off. I encourage the authors, if they have time, to explore simple paraphrasing-based DP methods, as they are relatively easy to implement and serve as strong defenses. A brief discussion on defenses, supported by some results, would greatly benefit readers seeking defense strategies, and if the authors provide such insights, I would be happy to change my rating to strong accept.

References:

[1] Privacy-and utility-preserving textual analysis via calibrated multivariate perturbations .WSDM 2020

[2] The Limits of Word Level Differential Privacy, EMNLP 2022.

[3] TEM: High Utility Metric Differential Privacy on Text, SIAM, 2023

[4] Locally differentially private document generation using zero shot prompting, EMNLP 2023

**Other Strengths And Weaknesses:**

Strengths:

- The work shifts the focus from traditional model-based MIAs to attacks based solely on synthetic data, addressing a realistic threat model where adversaries do not have direct access to the fine-tuned model.
- By proposing specialized canaries that blend an in-distribution prefix with a high-perplexity suffix, the authors enhance the detection capability of data-based MIAs, making privacy auditing more effective.


Weaknesses:

- A minor limitation is that the experiments are conducted on only two datasets, which may not capture the full diversity of real-world scenarios. The effectiveness of the proposed techniques across different domains or larger-scale datasets remains to be validated.

**Questions For Authors:**

See above

**Relation To Broader Scientific Literature:**

The authors examine data-driven MIA, proposing a fresh framework that offers a more realistic assessment of threats compared to model-based MIA.

**Theoretical Claims:**

NA

---

> ### Author Rebuttal · Authors · 2025-04-01
>
> We thank the reviewer for their feedback. We provide responses for the concerns raised below.
>
> > (1) A minor limitation is that the experiments are conducted on only two datasets
>
> We provide results for the *n*-gram based MIA for the SNLI dataset (for the setup from Table 1) below and will include this in the paper. These results suggest the same trends we report carry over to other datasets.
>
> |Canary injection|||||
> |-|-|-|-|-|
> |||AUC|TPR\@0.01|TPR\@0.1|
> |In-distribution||0.718|0.122|0.443|
> |Synthetic|Natural|0.534|0.016|0.111|
> ||Artificial|0.718|0.061|0.412|
>
> > (2) How do attacks work on differentially private text generation?
>
> We will add a section on mitigations strategies for our novel MIAs, focusing on fine-tuning the target model with DP-SGD before generating synthetic data, as in prior work (Yue et al., 2023; Mattern et al., 2022; Kurakin et al., 2023). Given past results (Table 3 in [1], Figure 3 in [2], or the results of the SaTML 2023 Membership Inference competition on STT-2 [3]), we expect the performance of model-based attacks to quickly decrease to a random guess baseline under DP guarantees. Since data-based attacks underperform compared to model-based attacks, and guarantees transfer to the synthetic data due to DP’s post-processing property, we expect data-based MIAs to also approach random guess for practical values of ε. By the end of the rebuttal phase, we aim to provide meaningful ablations on MIAs against DP-synthetic data, which we will then include in the paper.
>
> We leave other defense strategies (e.g. using paraphrasing techniques) as proposed by the reviewer for future work, and will elaborate on this in the discussion section.
>
> [1] Xie, et al. Differentially Private Synthetic Data via Foundation Model APIs 2: Text
>
> [2] Ma et al. Efficient and Private: Memorisation under differentially private parameter-efficient fine-tuning in language models.
>
> [3] Microsoft Membership Inference Competition (https://github.com/microsoft/MICO).

---

### Official Review · Reviewer_qpCC · 2025-03-16

**Overall Recommendation:** 3

**Summary:**

This paper proposes to audit the privacy risks from synthetic data generated by LLMs as synthetic data is becoming increasingly prevailing in different applications. The authors found that the typical canaries designed for model-based auditing were not effective for auditing the synthetic data. The paper proposed a new design for canaries that is better suited for auditing the data-based scenario. The method is analyzed empirically on benchmark datasets with various evaluation metrics.

**Claims And Evidence:**

The claims are supported by the evidence under the assumptions that the author made.

**Essential References Not Discussed:**

N/A

**Experimental Designs Or Analyses:**

The experiment designs are thorough. Though could be improved by analyzing how different domains might impact the auditing efficiency, e.g. would synthetic data generated for different domains in AG news make any difference in the auditing performance.

**Methods And Evaluation Criteria:**

The evaluation criteria makes sense to demonstrate the improvement in the auditing performance.

**Other Comments Or Suggestions:**

N/A

**Other Strengths And Weaknesses:**

Strengths:
1. The paper is considered a novel auditing scenario, as private synthetic data generation is more and more popular, understanding the privacy leakage from synthetic data is critical.
2. The paper is well-written with clear methodology, and thorough analysis on why the existing canaries failed and how to craft canaries for synthetic data auditing.

Weaknesses:
1. The evaluation can be strengthened by measuring the leakage against privacy-preserving methods such as private evolution, and DP fine-tuning for synthetic data generation.
2. The motivation for using n-gram for data-based attacks is not clearly described. Why n-gram is preferred, why not consider training a small neural network based model on synthetic data? How is n chosen?

**Questions For Authors:**

1. How does the size of synthetic data impact the auditing?
2. For similarity scores based on embeddings, how do different embedding models impact the auditing?

**Relation To Broader Scientific Literature:**

The paper is related to better understanding of privacy leakage through synthetic data which is a novel and important topic in the community.

**Theoretical Claims:**

N/A

---

> ### Author Rebuttal · Authors · 2025-04-01
>
> We thank the reviewer for their feedback. We provide responses for the concerns raised below.
>
> > (1) analyzing how different domains might impact the auditing efficiency
>
> In Table 1, we also study the effect of which labels (or domains) the canaries belong to. In particular, we consider both ‘natural’ and ‘artificial’ labels associated with canary samples, where ‘natural’ corresponds to labels from the same distribution as the labels from the original dataset and ‘artificial’ corresponds to a new, canary-specific label (see Sec. 4). We observe a slight increase in MIA performance across all data-based MIAs, suggesting that more rare, potentially artificially crafted labels make canaries more vulnerable. We leave a more thorough study on this effect to future work and will add this to the discussion section.
>
> > (2) measuring the leakage against privacy-preserving methods
>
> We will add a section on mitigations strategies for our novel MIAs, focusing on fine-tuning the target model with DP-SGD before generating synthetic data, as in prior work (Yue et al., 2023; Mattern et al., 2022; Kurakin et al., 2023). Given past results (Table 3 in [1], Figure 3 in [2], or the results of the SaTML 2023 Membership Inference competition on STT-2 [3]), we expect the performance of model-based attacks to quickly decrease to a random guess baseline under DP guarantees. Since data-based attacks underperform compared to model-based attacks, and guarantees transfer to the synthetic data due to DP’s post-processing property, we expect data-based MIAs to also approach random guess for practical values of ε. By the end of the rebuttal phase, we aim to provide meaningful ablations on MIAs against DP-synthetic data, which we will then include in the paper.
>
> [1] Xie, et al. Differentially Private Synthetic Data via Foundation Model APIs 2: Text
>
> [2] Ma et al. Efficient and Private: Memorisation under differentially private parameter-efficient fine-tuning in language models.
>
> [3] Microsoft Membership Inference Competition (https://github.com/microsoft/MICO).
>
> > (3) motivation for using n-gram for data-based attacks is not clearly described
>
> Many thanks for pointing this out. We have opted to train an n-gram model rather than a small, transformer-based model due to its simplicity and computational cost. Indeed, training the n-gram model on the synthetic data, both for SST-2 and AgNews takes less than 1 CPU minute. We further provide ablations of the value of n in Appendix H and Table 10, where we consistently find n=2 to be the optimal value. We will add the suggestion of training a small neural network instead to the discussion section.
>
> > (4) How does the size of synthetic data impact the auditing?
>
> We provide ablations for different sizes of the synthetic data in Appendix H and Figure 5. While we observe an improved performance for the n-gram based MIA as more synthetic data is generated, we maintain our main analysis considering a synthetic dataset of equal size to the private dataset as this is more realistic and used in prior work (Yue et al., 2023; Mattern et al., 2022; Kurakin et al., 2023).
>
> >  (5) How do different embedding models impact the auditing?
>
> For similarly-based methods, we opted for paraphrase-MiniLM-L6-v2 from sentence-transformers as the embedding model, as it offers great performance in semantic search benchmarks [link](https://www.sbert.net/docs/sentence_transformer/pretrained_models.html). As the performance of the MIA based on semantic similarity using this embedding model is outperformed by all other data-based MIAs (Table 1), we did not further ablate the choice of the embedding model. We leave this to future work and will add this to the discussion section.

---

### Official Review · Reviewer_L9fq · 2025-03-22

**Overall Recommendation:** 2

**Summary:**

This paper aims at investigating the privacy risks of synthetic text generated by LLMs by developing a new membership inference attack (MIAs). The main novelty of the proposed approach is that the adversary model considered only has access to the synthetic text generated by the model and not the model itself. Two MIAs are proposed for this setting.

**Claims And Evidence:**

Overall, the claims with respect to the difference between model-based and data-based MIAs are supported only through experiments on two datasets, which raises some doubts on whether the observed results will carry to other datasets. However, a large set of variations of these experiments have been conducted, which demonstrate the robustness of the proposed approach.

**Essential References Not Discussed:**

Essential references, including recent ones, have been cited in the paper.

**Experimental Designs Or Analyses:**

Overall, the experimental evaluation is well-explained and seems sound. There is however no justification of the choice of parameters that have been used for the two variants of the MIA proposed. The paper also lacks experiments for evaluating how the success of the proposed attack will fare against differentially-private variant of the model training.

**Methods And Evaluation Criteria:**

The difference between model-based attacks and data-based attacks are not well explained. The process for generating a synthetic text is also unconventional as it seems to indicate that the objective of the model is to create texts that match desired labels while in general the text will be generated based on the indication of the prompt. In addition, the fact that the synthetic dataset should be the same size as the original dataset is not very realistic are most LLMs are usually trained on a very large corpus (e.g., possibly the whole Internet). This major issue should at least be acknowledged and discussed in the paper.

**Other Comments Or Suggestions:**

Figure 5 in the supplementary material has some issues with the corresponding legends.

**Other Strengths And Weaknesses:**

The main novelty of the proposed approach is the adversary model considered, which only leverages the synthetic produced, and proposes two MIAs for this setting.

**Questions For Authors:**

Do you have any way to verify if the two datasets considered for the experiments were not part already of the training set of the LLM considered?

**Relation To Broader Scientific Literature:**

The authors have done a good job at reviewing previous works on membership inference attacks against LLMs and synthetic tabular data. The proposed approach is also well-situated compared to existing works although the adversary model considered is non-standard.

**Theoretical Claims:**

Currently, the authors do not discuss the possibility that the training set and the canaries have been possibly used during the training of the LLM rather than simply during the fine-tuning. In particular as the Stanford Sentiment Treebank and the AG news datasets have respectively been published in 2013 and 2015, there is a high chance that they have been seen in the training set of the LLM models considered.

---

> ### Author Rebuttal · Authors · 2025-04-01
>
> Thanks for the feedback; we provide detailed responses below.
>
> > Difference between model-based and data-based MIAs supported only through experiments on two datasets. A large set of variations of these experiments have been conducted, which demonstrate the robustness of the proposed approach.
>
> We are glad the reviewer thinks our experiments demonstrate the robustness of our approach. We chose to investigate the gap between model- and data-based attacks in depth through detailed ablation studies rather than shallowly on a broader range of datasets.
>
> We provide results for the *n*-gram based MIA on SNLI (same setup as Table 1), which we will include in a revision. Results suggest the same trends we report carry over to other datasets.
>
> |Canary injection|||||
> |-|-|-|-|-|
> |||AUC|TPR\@0.01|TPR\@0.1|
> |In-distribution||0.718|0.122|0.443|
> |Synthetic|Natural|0.534|0.016|0.111|
> ||Artificial|0.718|0.061|0.412|
>
> > Difference between model-based and data-based attacks not well explained
>
> We thoroughly describe the difference between model- and data-based MIAs in Sec. 2, including pseudocode in Alg. 1. Appendix E contains additional discussion on the difference between the threat models. We present concrete model- and data-based attacks in Sec. 3.1, including the calculation of membership inference signal and our adaptation of RMIA. We detail the choice of hyperparameters and how we evaluate attacks in practice in Sec. 4. We include additional details about model- and data-based attacks in Appendices A, B.
>
> We welcome suggestions to make this clearer if what we already provide is not enough.
>
> > The process for generating a synthetic text is unconventional
>
> We generate synthetic data as done conventionally (Yue et al., 2023; Mattern et al., 2022; Kurakin et al., 2023), by prompting the model on label-dependent templates (cf. Appendix C), so that text is generated based on the indication of the prompt.
>
> > The fact that the synthetic dataset should be the same size as the original is not realistic
>
> We report on experiments on synthetic datasets 2-8x larger than the original in Appendix I.
>
> We focus on generating synthetic data derived from a private dataset so that e.g. downstream tasks on the synthetic dataset have similar utility. Hence, in our main experiments we generate datasets matching the size and label histogram of the private data. This is the same setting studied by Yue et al., 2023; Mattern et al., 2022; Kurakin et al., 2023.
>
> > Possibility that the training set and the canaries have been used during pre-training
>
> Mistral-7B's training data is not public, so we cannot rule out SST-2/AG News being included in the training data. However, we aim to measure the difference in performance between model- and data-based MIAs on the fine-tuning dataset of a model used to synthesize data, which training and fine-tuning data overlap would affect similarly.
>
> We can, however, rule out the presence of canaries in the training data when they are constructed artificially (in-distribution prefix F=0; Table 1, Fig. 1(a,b,d,e), Fig. 2). Especially at high perplexities, these canaries are likely absent from the training data. For canaries with F>0 and if parts of the datasets were included in pretraining, it would, if anything, make MIAs more challenging. We will include this discussion in an eventual revision.
>
> > No justification of the choice of parameters used for the two variants of the MIA proposed.
>
> We discuss hyperparameter selection in Appendix H (paragraph “Hyperparameters in data-based attacks”). We consistently find the best performance for *n*=2 (*n*-gram MIA) and *k*=25 (number of closest synthetic records for similarly based MIAs), which we used in the main experiments.
>
> > differentially-private variants of the model training.
>
> We will add a section on mitigations, focusing on fine-tuning the target model with DP-SGD before generating synthetic data as in prior work (Yue et al., 2023; Mattern et al., 2022; Kurakin et al., 2023). Given past results (Table 3 in [1], Figure 3 in [2], or the results of the SaTML 2023 Membership Inference competition on SST-2 [3]), we expect performance of model-based attacks to decrease to a random guess baseline under DP. Since data-based attacks underperform compared to model-based attacks and guarantees transfer to synthetic data due to DP’s post-processing property, we expect data-based MIAs to also approach random guess for practical values of ε. By the end of the rebuttal phase, we aim to provide meaningful ablations on MIAs against DP-synthetic data, which we will then include in the paper.
>
> [1] Xie, et al. Differentially Private Synthetic Data via Foundation Model APIs 2: Text
>
> [2] Ma et al. Efficient and Private: Memorisation under differentially private parameter-efficient fine-tuning in language models.
>
> [3] Microsoft Membership Inference Competition (https://github.com/microsoft/MICO).
>
> > Fig.5 legibility.
>
> Thanks for pointing this out, we will address this.

---

### Decision · Program_Chairs · 2025-05-01

**Decision:**

Accept (poster)

**Comment:**

This paper studies the privacy risk of synthetic data generation. To this end, this paper proposes novel methods for generating canaries and testing for their leakage.

Overall, the reviewers agreed that this was an interesting setting with good empirical evaluation. The results show promise and the paper was well-written.

However, there were also a few concerns that reviewers noted:
* Lacking justification for and explanation of the n-gram attack
* Some issue with the experimental design and # of datasets considered
* Possibly other attacks
* Considering models trained with DP.

Some of these were addressed within the rebuttal. Overall, the paper appears to be have improved during the rebuttal and provides a good contribution.